# Promoter-proximal elongation regulates transcription in archaea

Fabian Blombach [1✉], Thomas Fouqueau[1], Dorota Matelska [1], Katherine Smollett [1] & Finn Werner [1✉]

Recruitment of RNA polymerase and initiation factors to the promoter is the only known target for transcription activation and repression in archaea. Whether any of the subsequent steps towards productive transcription elongation are involved in regulation is not known. We characterised how the basal transcription machinery is distributed along genes in the archaeon *Saccharolobus solfataricus*. We discovered a distinct early elongation phase where RNA polymerases sequentially recruit the elongation factors Spt4/5 and Elf1 to form the transcription elongation complex (TEC) before the TEC escapes into productive transcription. TEC escape is rate-limiting for transcription output during exponential growth. Oxidative stress causes changes in TEC escape that correlate with changes in the transcriptome. Our results thus establish that TEC escape contributes to the basal promoter strength and facilitates transcription regulation. Impaired TEC escape coincides with the accumulation of initiation factors at the promoter and recruitment of termination factor aCPSF1 to the early TEC. This suggests two possible mechanisms for how TEC escape limits transcription, physically blocking upstream RNA polymerases during transcription initiation and premature termination of early TECs.

[1] Division of Biosciences, Institute of Structural and Molecular Biology, University College London, London, UK. ✉email: f.blombach@ucl.ac.uk; f.werner@ucl.ac.uk

The rate of RNA synthesis is determined by the frequency of transcription initiation and premature termination[1]. The recruitment stage of transcription initiation is the main target for regulation in yeast and bacteria[2,3], however, the initiation rate can also be affected indirectly by downstream events. In metazoans, promoter-proximal pausing of RNA polymerase (RNAP) II with slow turnover times blocks pre-initiation complex (PIC) formation for the following RNAP and is thereby widely rate-limiting for transcription initiation[4,5]. Promoter-proximal paused RNAPII can also be subject to premature termination providing an additional way of transcription regulation[6–10]. Promoter-proximal RNAP dynamics also limit gene expression in E. coli[11]. Well-established processes of post-recruitment regulation include Sigma70-dependent pausing and transcription attenuation mediated by premature termination[12–14]. Another possible underlying molecular mechanism might be pausing during initial transcription[15,16], though its contribution to genome-wide gene regulation remains to be investigated[17].

Archaea form the 'third domain of life' next to bacteria and eukaryotes, with the latter likely originating from an archaeal ancestor[18]. The basal archaeal transcription machinery represents an evolutionarily ancient core of the RNAPII system encompassing RNAP subunits, basal transcription initiation and -elongation factors and core promoter elements[19–21]. The mechanisms of initiation have been characterised in great detail in vitro (Fig. 1a). The basal transcription factors TBP and TFB bind to their cognate promoter elements (TATA box and BRE, respectively) and sequester RNAP to form the minimal PIC[22,23]. A third transcription initiation factor TFE binds to RNAP to form the complete PIC and facilitates DNA melting leading to formation of the open complex[24–27]. TFE stimulates transcription initiation but is not strictly required in vitro. Like the PIC, the transcription elongation complex (TEC) corresponds to an evolutionarily ancient RNAPII TEC encompassing homologues of a subset of RNAPII elongation factors: Spt4/5 (DSIF in human)[28,29] and potentially the archaeal homologue of elongation factor Elf1 (Elof1 in humans)[30,31]. In addition, the transcript cleavage factor TFS (homologous to TFIIS) transiently associates with the TEC and reactivates arrested TECs[32]. Spt4/5 and TFE bind to RNAP in a mutually exclusive manner and the transition from transcription initiation to elongation requires factor switching between TFE and Spt4/5[33]. Transcription termination in archaea occurs via intrinsic or factor-dependent mechanisms. The latter involves termination factor aCPSF1 (or FttA)[34], a ribonuclease that is evolutionary related to the RNAP II termination factor CPSF73 and the integrator subunit Ints11.

Archaeal promoters seem to comprise fewer promoter elements compared to their bacterial and eukaryotic counterparts, but it is possible that additional unknown sequence elements as well as the physicochemical properties of promoter DNA contribute to promoter strength[35,36]. Likewise, our understanding of transcription regulation is limited to factors modulating the recruitment of PICs[37,38] where repression generally involves steric hindrance of RNAP or basal initiation factor binding and activation is achieved by enhancing their binding[39–42]. How archaeal RNA polymerase progresses further through the transcription cycle and whether subsequent stages beyond initiation are targeted for transcription regulation in archaea is currently poorly understood.

The crenarchaeon Saccharolobus solfataricus (formerly Sulfolobus) is a well-established model organism for archaeal transcription (e.g., refs. [24,32,43–45]). Importantly, S. solfataricus harbours the full repertoire of known archaeal transcription initiation and elongation factors including Elf1 making it a good model to study mechanisms of transcription regulation beyond initiation.

We analysed the genome-wide distribution of RNAP and transcription initiation and elongation factors in S. solfataricus by using a multi-omics approach including chromatin immunoprecipitation-sequencing-based techniques (ChIP-seq) and transcriptomics. Our results provide evidence for a sequential recruitment cascade of elongation- (Spt4/5 and Elf1) and termination- (aCPSF1) factors to RNAPs in the promoter-proximal region of the transcription unit. We show that escape of TECs from this region is rate-limiting for transcription and subject to regulation. Thereby we establish early elongation as an important checkpoint to set and regulate promoter strength in archaea.

## Results

**Uniform PIC assembly during exponential growth.** We mapped the genome-wide occupancy of RNAP, initiation-, elongation- and termination factors to shed light on how the individual stages of transcription are subject to transcription regulation in S. solfaticus. We developed and adapted ChIP-seq using polyclonal antibodies raised against RNAP subunits Rpo4/7 and recombinant transcription factors. In order to obtain the resolution that separates PICs from promoter-proximal, early TECs, we adapted a ChIP-exo approach for RNAP and initiation factors TFB and TFEβ that includes 5′->3′ exonuclease-trimming of the immunoprecipitated-DNA fragments[46].

We calculated aggregate profiles of ChIP-exo data for a set of 298 TUs with mapped TSSs. These TUs were selected from a total set of 1054 mapped TSSs based on the corresponding ChIP-seq data for TFB and TFEβ to include only TUs displaying TFB and TFEβ occupancy as well as to exclude TUs with problematic regions for the mapping of sequencing reads (see Methods section). The profiles showed a distinct footprint for RNAP and initiation factors TFB and TFEβ around the TSS (Fig. 1b). The overall similarity of the RNAP, TFB and TFEβ profiles reflect the footprints of entire cross-linked PICs rather than the DNA-binding sites of the individual factors within the PIC. The main upstream border of the PIC is formed by a broad peak centred around position −12 to −14 that can be most likely attributed to the N-terminal cyclin fold of TFB interacting with the DNA downstream of the TATA-box, which is in good agreement with ChIP-exo mapping of RNAPII PICs[5,47]. The downstream border for the PIC signal on the template strand was relatively broad and reached well beyond the ~20 bp downstream of the TSS protected in in vitro exonuclease foot-printing experiments of archaeal PICs[48,49] (Fig. 1b).

To investigate any heterogeneity in the recruitment of basal factors, we quantified the ChIP-exo signal on the non-template strand over a window from −30 to +20 relative to the TSS (Fig. 1c). Both TFB- and TFE occupancy correlated strongly with RNAP (Spearman's $r = 0.92$ in both cases, Fig. 1d, e). Since TFB binding is critically dependent on TBP binding, and TFE binding depends on RNAP, our results suggest that all components of the archaeal PIC (TBP, TFB, RNAP and TFE) assemble on promoters in a homogenous, or uniform, fashion.

We expected exceptions to this rule where transcription regulators would interfere with the recruitment of RNAP. E.g., the SSO8620 promoter shows strong TFB- but weak RNAP- and TFEβ signals, which indicates repression of RNAP recruitment to the TBP-TFB ternary complex. Consistent with this notion, the predominant TFB footprint on SSO8620 was significantly narrower compared to TFB footprints on promoters showing unimpaired RNAP recruitment to the PIC (Supplementary Fig. 1).

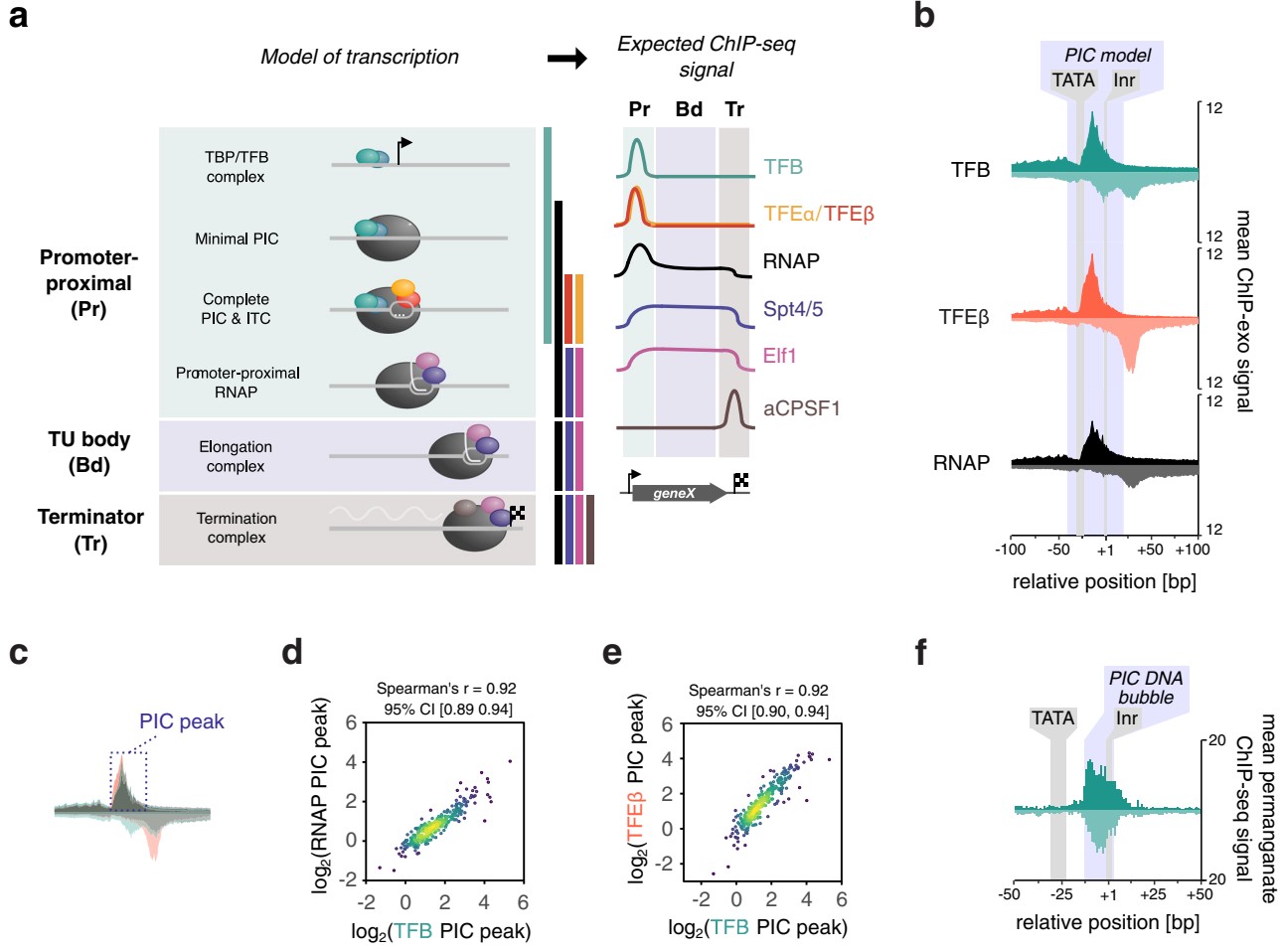

**Fig. 1 Uniform PIC assembly during exponential growth. a** Stages of transcription in archaea can be assessed by a combination of RNAP and basal transcription initiation and elongation factor occupancy. **b** Aggregate plots of ChIP-exo signal at the promoter for RNAP and initiation factors ($n = 298$ TUs). The average signal on the non-template and template strand is shown above and below the line, respectively. Data are pooled from three biological replicates. **c** Schematic showing the 50-nt window on the non-template strand that we used to quantify and correlate the ChIP-exo signal. Because RNAP and initiation factors TFB and TFEβ yield similar profiles on the non-template strand, this signal can be attributed to the PIC. **d**, **e** Scatter plots depicting correlation between the main ChIP-exo signal for TFB and RNAP (**d**) or TFEβ (**e**) within the PIC ($n = 298$). Data represent the average signal over a 50-nt window that we attributed to the PIC (see panel **c**). The geometric mean of three biological replicates is shown. **f** Aggregate plots of permanganate ChIP-seq signal at the promoter for initiation factor TFB ($n = 298$ TUs). The average signal for T-encoding positions on the non-template and template strand is shown above and below the line, respectively. Data are pooled from two biological replicates.

One possible explanation for why the ChIP-exo footprint of PICs was extended downstream could be that the PICs might be in a state of extended DNA scrunching where they 'reel in' downstream DNA during initial transcription[50]. DNA scrunching results in downstream extension of the DNA bubble thereby making thymine bases within the melted region sensitive to permanganate. Promoter clearance by RNAP limits the extent of DNA scrunching and in vitro crosslinking data suggest that archaeal RNAP clears from the promoter approximately when it reaches position +10[51]. To test whether PICs undergo extended DNA scrunching in vivo beyond the anticipated position of promoter clearance, we mapped the melted DNA regions in the PIC genome wide by permanganate ChIP-seq using TFB as IP target[52,53]. Aggregate plots showed that DNA melting occurred in the −12 to +3 region relative to the TSS, peaking at position −10 (Fig. 1f), which is consistent with the in vitro permanganate foot printing of reconstituted PICs[24,49,54]. Importantly, the signal decreased to background levels beyond position +10, the expected point of promoter clearance. Thus, extended DNA scrunching is unlikely to explain the downstream border of PICs.

The discrepancy between in vitro exonuclease and in vivo ChIP-exo footprints suggest that additional, yet uncharacterised components associate with the PIC in the cell such as chromatin proteins or extended interaction of the PIC with downstream DNA.

**RNAP escape limits productive transcription.** How does archaeal RNAP progress from transcription initiation into productive elongation? To address this poorly understood process, we generated paired-end ChIP-seq data sampled to a mean fragment size of 120 bp. Because the *S. solfataricus* genome has very short intergenic regions with juxtaposed promoters of different TUs, ChIP-seq data with such short mean fragment size provide a good compromise between the requirement of good spatial resolution and the overall higher robustness of ChIP-seq compared to ChIP-exo. These data ensured unequivocal assignment of initiation factor peaks to specific promoters. At the promoters, the RNAP, TFB and TFEβ ChIP-seq data were in good agreement with ChIP-exo data, i.e., indicative of uniform PIC assembly (Supplementary Fig. 2). Choosing transcription

units with lengths of >500 bp provided us with a window within the TU body where the RNAP occupancy reflects the productive elongation phase well-separated from the PIC signal. Crucially, some TUs showed a strong decrease in RNAP occupancy from the promoter towards the TU body revealing heterogeneity in how RNAPs progress into productive elongation (Fig. 2a). For TUs encoding housekeeping genes such as *thsB* (encoding a subunit of the thermosome chaperone, Fig. 2b), or *rps8E* (ribosomal protein S8e) (Fig. 2c) the decrease in RNAP occupancy was rather small. In contrast, for example *dhg-1* — one of two glucose-1-dehydrogenase isoenzymes — shows a drastic decrease in RNAP occupancy (Fig. 2d). Notably, all CRISPR arrays showed strongly reduced RNAP escape into productive transcription suggesting that crRNA synthesis is regulated at this level (Fig. 2e). If the transition into the productive transcription elongation phase is rate-limiting for RNA synthesis genome-wide, global mRNA levels should correlate better with RNAP occupancy within the TU body than RNAP occupancy at the promoter. Consistent with the rate-limiting role of RNAP escape, mRNA expression levels of the first cistron in the TU did correlate significantly better with the average RNAP occupancy within the TU body ($RNAP_{Bd}$, calculated over positions $+251$ to $+500$) than with RNAP promoter occupancy ($RNAP_{Pr}$. Spearman's $r$ of 0.75 versus 0.44, Fig. 2f, g), or indeed TFB occupancy at the promoter (Spearman correlation statistically not significant, Fig. 2h).

In summary, the ChIP-seq results reveal that the escape of RNAP into productive elongation varies greatly across different TUs.

**TECs accumulate in the promoter-proximal region**. The observed accumulation of RNAP in the promoter-proximal region can be due to PICs or TECs. If TECs accumulate in this region, then the elongation factor Spt4/5 and possibly Elf1 should show similar promoter-proximal accumulation as RNAP. To test this, we classified TUs based on their RNAP escape by calculating an escape index (EI) for each TU defined as the log-transformed ratio of $RNAP_{Bd}$ over $RNAP_{Pr}$. We divided the TUs into two subsets with a high (EI $> -1$) or low escape index (EI $< -2.5$) and compared the aggregate profiles for RNAP, Spt4/5 and Elf1 (Fig. 3a, b). Both elongation factors accumulate in the promoter-proximal region of TUs with low EI alongside RNAP (Fig. 3b). In support of this, escape index calculations for both elongation factors revealed strong correlations with RNAP EI (Supplementary Fig. 3). Escape indices for all three proteins (RNAP, Spt4/5 and Elf1) were strongly correlated with mRNA expression levels (Supplementary Fig. 4). Thus, the observed accumulation of RNAP in the promoter-proximal region appears to reflect reduced TEC escape into productive transcription.

Notably, Spt4/5 is consistently recruited to the TEC prior to Elf1, independent of whether escape is high or low (Fig. 3a, b). Consecutive recruitment of elongation factors is consistent with the observation that some TUs with low RNAP escape showed a much stronger Spt4/5 recruitment compared to Elf1 suggesting that TECs stall at an earlier stage, prior to Elf1 recruitment, on these TUs (Supplementary Fig. 5). Henceforth we refer to the two TEC complexes as $TEC_{Spt45}$ and $TEC_{Spt45-Elf1}$. Our data thereby also provide the first experimental evidence that the archaeal Elf1 homologue is a general part of the archaeal TEC.

In order to corroborate the differences in TEC escape with a second, independent method, we characterised the nascent RNAs synthesised at the 5′ end of the TUs referred to as TSS-RNAs. Short RNAs (20 to 200-nt length) were isolated, enriched for triphosphorylated 5′-ends using the Cappable-seq method[55] and deep-sequenced. In quantitative terms, the occupancy of promoter-proximal elongation complexes (using $Spt4/5_{Pr}$ as

proxy) correlated well with TSS-RNA read counts (Spearman's $r = 0.61$) (Fig. 3c) and significantly better than mRNA counts from total RNA-seq (Spearman's $r = 0.48$) (Supplementary Fig. 6). This is in line with the isolated short 5′ triphosphorylated RNAs being predominantly composed of nascent RNAs synthesised by early TECs rather than degradation products of full-length RNAs. In qualitative terms, TUs with low RNAP escape were associated with the synthesis of shorter TSS-RNAs (<50 nt) consistent with promoter-proximal accumulation of TECs (Fig. 3d).

In summary, our results demonstrate that RNAP accumulates in the promoter-proximal region in the form of early TECs that incorporate Spt4/5 and Elf1.

**aCPSF1 recruitment to the TEC correlates with reduced TEC escape**. The balance between premature termination and antitermination in the promoter-proximal regions of genes is a well characterised mode of transcription regulation in bacteriophages and bacteria[56], and has more recently also been reported for eukaryotic transcription systems[57]. Premature termination could also contribute to the observed promoter-proximal enrichment of TECs that we observed in archaea. As we found no evidence for any significant sequence bias in the promoter-proximal region including uridine-stretches that could serve as intrinsic terminators, we considered factor-dependent termination mediated by the archaeal termination factor aCPSF1[34,58,59]. aCPSF1 is capable of inducing transcription termination on TECs stalled in the promoter-proximal region ($+54$)[34]. Intriguingly, aCPSF1 accumulated in the promoter-proximal-region of most TUs (188 out of 212 TUs with peaks passing detection threshold) including *thsB*, *rps8E* and *dhg-1* (Fig. 2b–d). We also analysed aCPSF1 association with predicted TSSs from pairs of TUs in divergent orientation. These TSSs are thus isolated from termination sites at the 3′-end of TUs. aCPSF1 peaks showed strong association with these TSSs similar to initiation factor TFB that we used as control in line with the results above (Supplementary Fig. 7).

In contrast to the promoter-proximal aCPSF1 peaks, aCPSF1 does not form clearly defined peaks at 3′-ends of most TUs predicted from RNA-seq data. Instead, we observed a decrease in occupancy of aCPSF1 together with RNAP downstream of the predicted mRNA 3′-ends, and only in some cases well-defined CPSF1 peaks (Supplementary Fig. 8).

Provided that the promoter-proximal occupancy reflects recruitment of aCPSF1 to TECs, the distribution of aCPSF1 in the promoter-proximal region should depend on the distribution of RNAP. Accordingly, the aCPSF1 peaks sharpened on TUs with low RNAP escape likely due to a lower elongation rate or processivity (Fig. 3a, b). aCPSF1-mediated transcription termination is stimulated in the presence of Spt4/5 in vitro suggesting that Spt4/5 might facilitate aCPSF1 recruitment to the TEC or modulate aCPSF1 activity[34]. In line with the in vitro observations, TUs with low RNAP escape recruited aCPSF1 consecutive to Spt4/5 (Fig. 3b).

Provided that aCPSF1 recruitment results in the premature termination of elongation complexes, its recruitment to the promoter-proximal TECs should decrease TEC escape and RNA levels. The CPSF1 recruitment was indeed inversely associated with TEC escape (Fig. 3e). This anticorrelation holds true whether the aCPSF1 load is calculated as ratio of aCPSF1 to Elf1 (Fig. 3e) or aCPSF1 to Spt4/5 promoter occupancy (Supplementary Fig. 9). Importantly, a higher aCPSF1 load was correlated with lower mRNA levels (Fig. 3f and Supplementary Fig. 9).

In summary, our data show that promoters with high levels of promoter-proximal aCPSF1 recruitment show decreased TEC escape and low mRNA levels. These observations demonstrate the

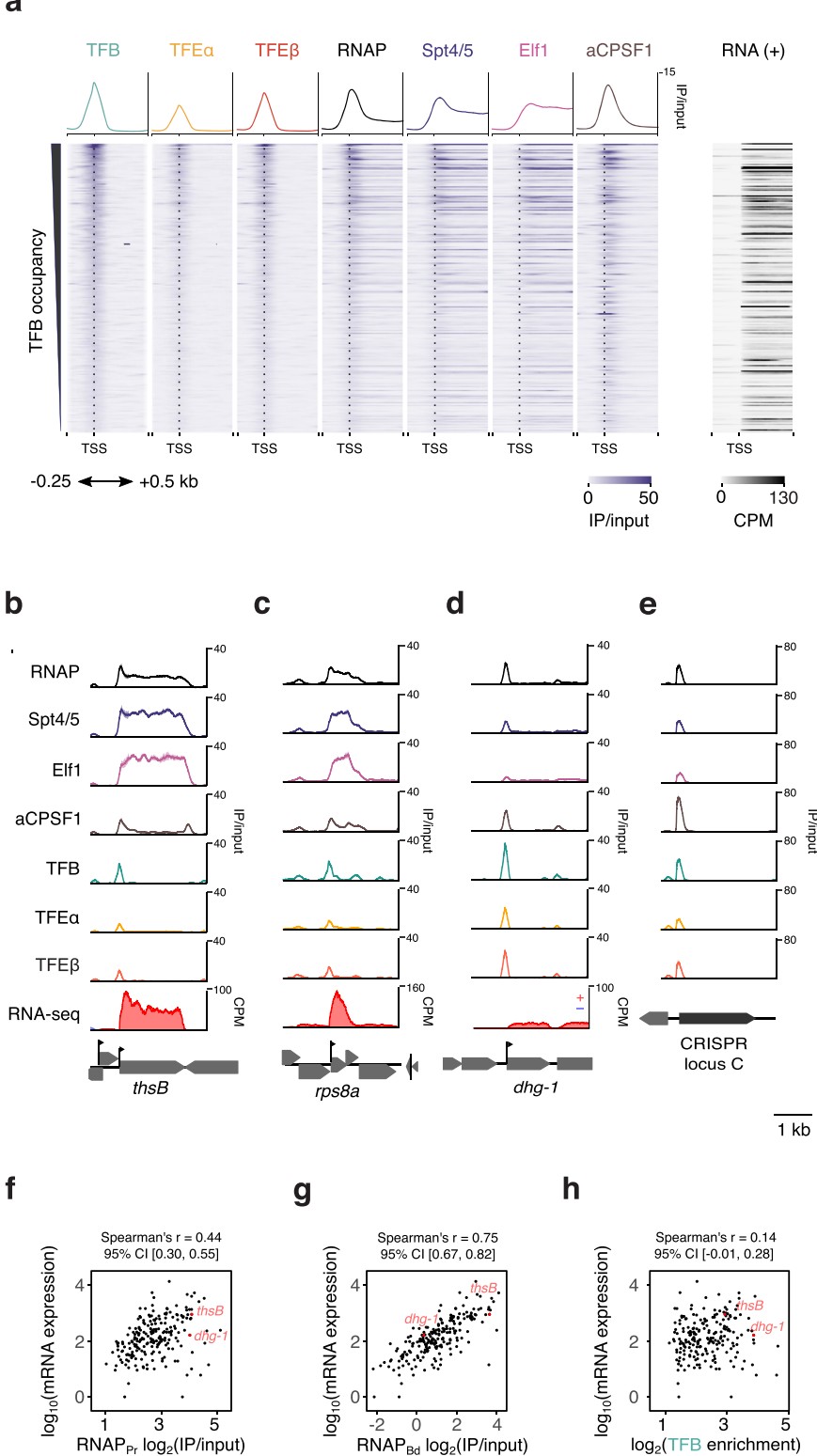

**Fig. 2 Productive transcription is limited by RNAP escape. a** Heatmap of ChIP-seq data for RNAP and the basal transcription machinery on a selected set of 212 TUs for exponential growth phase. The corresponding RNA-seq data for the plus strand are depicted on the right. Data are based on one representative of two biological replicates. **b**–**e** ChIP-seq occupancy plots on *thsB* coding for a subunit of the thermosome chaperone complex (**b**), *rps8e* (**c**), *dhg-1* coding for a glucose-1-dehydrogenase (**d**) and *CRISPR C* (**e**). Traces show mean occupancy for two biological replicates with the range depicted as semi-transparent ribbon. **f**–**h** Correlation of steady-state mRNA levels with RNAP occupancy at the promoter (RNAP$_{Pr}$, **f**), the TU body (RNAP$_{Bd}$, **g**) and TFB promoter occupancy (**h**), $n = 211$ TUs. ChIP-seq data represent the geometric mean of two biological replicates. Rockhopper[88] estimates of mRNA levels are based on two biological replicates.

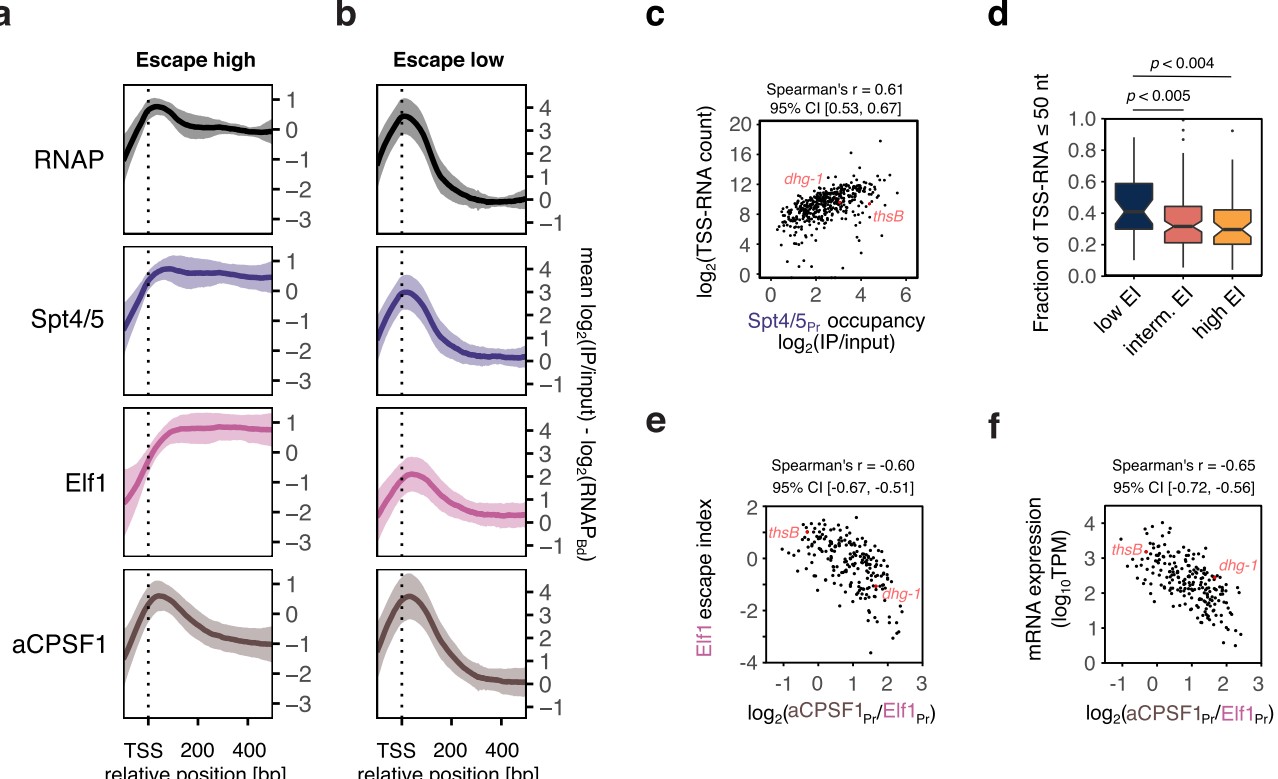

**Fig. 3 Promoter-proximal elongation determines RNAP escape. a**, **b** Recruitment of elongation factor Spt4/5 precedes Elf1 and aCPSF1 recruitment to the TEC. Aggregate plots for TUs with high (EI RNAP >−1, $n = 58$) (**a**) and low escape indices (<−2.5, $n = 41$) from the set of 212 TUs analysed in Fig. 2 (**b**). Before averaging across TUs, RNAP, Spt4/5, Elf1 and aCPSF1 ChIP-seq occupancy was scaled to the average RNAP occupancy within the body of each specific TU (RNAP$_{Bd}$). The profiles thus represent the relative recruitment of the factors to the TEC. Lines represent mean values across TUs with 1x standard deviation shown as semi-transparent ribbon. Data are from a single representative of two biological replicates. **c** Scatter plot depicting the correlation between Spt4/5$_{Pr}$ and TSS-RNA ($n = 438$). Data represent the mean of two biological replicates. **d** TSS-RNA length distribution correlates with TEC escape. Boxplots depicting the fraction of TSS-RNAs smaller than 50 nt for TUs with low, intermediate, and high RNAP escape. Boxplots represent median, interquartile ranges and Tukey-style whiskers. Statistical significance of the observed differences was tested using a one-sided wilcoxon rank sum test. **p < 0.01. The number of TUs in each EI category was 35 (low EI), 109 (intermediate EI) and 56 (high EI). Data represent the mean of two biological replicates. **e** Scatter plots depicting the anticorrelation between Elf1 EI (mean of two biological replicates) and the relative load of aCPSF1 on the Elf1-bound TEC calculated as aCPSF1$_{Pr}$ to Elf1$_{Pr}$ ratio (geometric mean of two biological replicates, $n = 212$ genes). **f** Correlation of aCPSF1 load (aCPSF$_{Pr}$/Elf1$_{Pr}$) to mRNA expression levels (mean of two biological replicates, $n = 211$ genes).

link between the termination factor aCPSF1 and RNA output and are consistent with a premature termination mechanism.

**The *CRISPR C* promoter shows pausing in the promoter-proximal region in vitro**. To test whether promoter-proximal pausing of RNAP can be observed in vitro for low TEC escape promoters, we developed a synchronised in vitro transcription assay to monitor promoter-proximal transcription elongation dynamics. We used *S. solfataricus* cell lysates generated from cells in exponential growth phase to provide a full set of auxiliary factors.

To generate templates for in vitro transcription, we inserted target promoter regions encompassing −50 to +100 relative to the TSS into plasmids that were subsequently linearised downstream of the insert to allow for run-off transcripts of 115 nt length. Synchronisation of transcription was achieved by a transcriptionally inhibitory variant of TFB termed TFBc that comprises only the C-terminal cyclin fold domains[60]. Pre-formed PICs are able to initiate a single round of transcription, but subsequent PIC assembly and transcription re-initiation is blocked by an excess of TFBc outcompeting the inherent TFB for recruitment to TBP-bound promoters (Fig. 4a). The generated transcripts were affinity purified using immobilised 25 nt

antisense oligonucleotides. We tested the assay on two promoters showing high TEC escape (*thsB* and *rps8e*) and two promoters showing low TEC escape (*dhg-1* and *CRISPR C*), the same promoters with ChIP-seq profiles depicted in Fig. 2b–e. All four promoters gave raise to run-off transcripts within 30 s under the experimental conditions (Fig. 4b). Notably, the *CRISPR C* promoter displayed some level of early, broad pausing at 30–40 nt transcript length, about the shortest length that is reliably detectable with the assay. Thus, the in vitro transcription data for the *CRISPR C* promoter are in line with early pausing of TECs in the promoter-proximal region. The absence of a corresponding pausing pattern for the *dhg-1* promoter may suggest that it is difficult to establish the proper context such as chromatinization of the DNA templates that reflects the in vivo situation.

**TEC escape regulation contributes to oxidative stress response**. Our results demonstrate that TEC escape is an important factor for determining promoter strength and RNA levels. In order to investigate whether cells can modulate TEC escape to regulate transcription, we tested how TEC escape changed in response to environmental changes such as oxidative stress. The impact of oxidative stress on *S. solfataricus* using a hydrogen peroxide

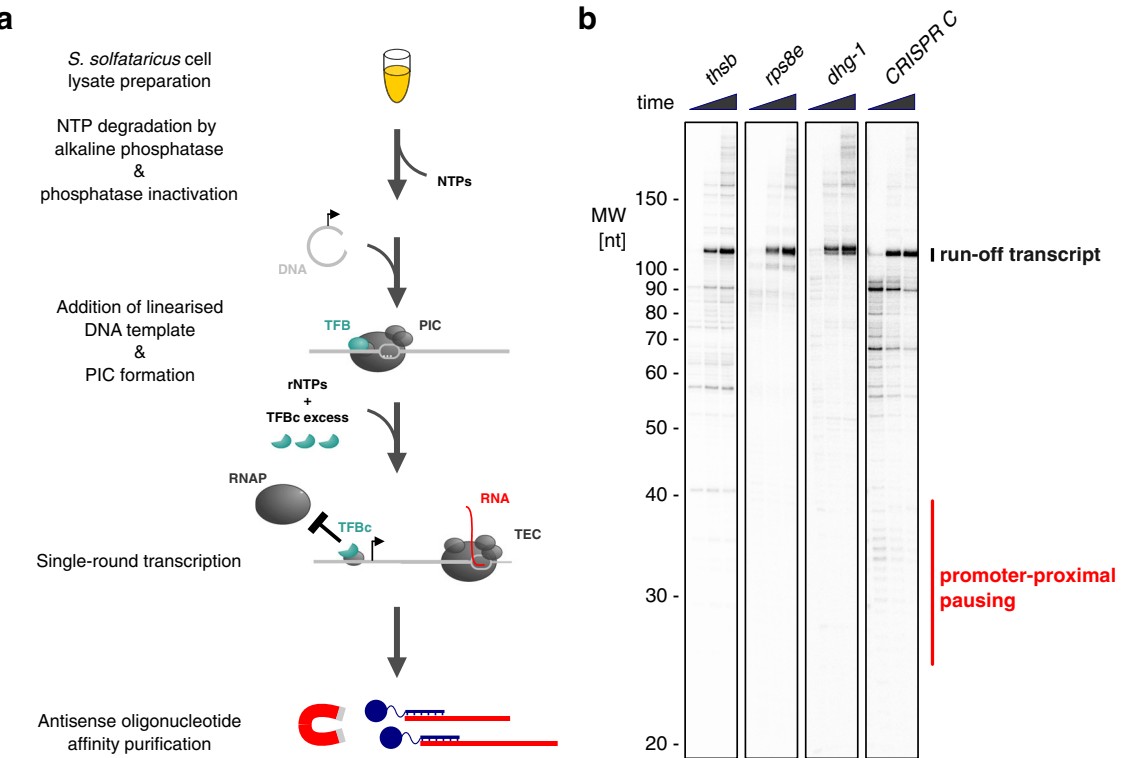

**Fig. 4 The *CRISPR C* promoter shows early pausing in vitro. a** Schematic overview of the cell lysate-based synchronised in vitro transcription system for *S. solfataricus*. Cell lysates were treated with shrimp alkaline phosphatase for NTP degradation before heat inactivation of the phosphatase. Linearised plasmid DNA containing a *S. solfataricus* promoter were added to allow PIC formation on the templates with inherent initiation factor TFB. Simultaneously with the addition of ribonucleotides to allow the PICs to initiate a single round of transcription, we added an excess of a recombinant TFB variant termed TFBc that blocks subsequent rounds of PIC formation. The generated transcripts were purified by affinity purification using immobilised 25 nt antisense oligonucleotides. **b** Synchronised in vitro transcription assay with two promoters showing high TEC escape (*thsB* and *rps8e*) and two promoters showing low TEC escape in vivo (*dhg-1* and *CRISPR C*). Samples were withdrawn 15 s, 30 s and 45 s after simultaneous addition of 50 μM rNTPs including [α-$^{32}$P]-UTP and TFBc. Purified radiolabelled transcripts were resolved on a denaturing polyacrylamide gel. The position of run-off transcripts and transcripts resulting from pausing in the promoter-proximal region is indicated. A representative experiment of three technical replicates is shown.

treatment protocol has been partially characterised[61]. We expected that besides the induction of transcription for stress genes such as *dps-l*[61], oxidative stress would cause a broader, global transcriptional response such as a widespread attenuation of the transcriptome. Relevant to transcription initiation, the peroxide treatment results in the depletion of the TFEβ-subunit from the cytoplasm (Supplementary Fig. 10). TFEβ depletion coincided with globally decreased promoter occupancies for both TFEα and TFEβ subunits in ChIP-seq implying that TFEα recruitment to the promoter is TFEβ-dependent (Supplementary Fig. 11).

To understand how oxidative stress affects TEC escape, we generated ChIP-seq data for RNAP, initiation- and elongation factors and we applied the same data filtering as for exponential growth phase data to obtain a set of 118 TUs with EI estimates. We then analysed the intersection of 71 TUs from both exponential growth and oxidative stress data sets to compare TEC escape between the two conditions. TUs displaying high TEC escape under exponential growth conditions showed overall reduced RNAP and Spt4/5 escape in response to oxidative stress (Fig. 5a). In addition, the promoter-proximal recruitment of aCPSF1 and the negative correlation to TEC escape are reduced in response to oxidative stress (Supplementary Fig. 12). The lower aCPSF1 signal cannot be explained by protein depletion because immunodetection revealed that the protein levels remained unaffected by oxidative stress (Supplementary Fig. 10).

The comparison between exponential growth and oxidative stress provides us with an opportunity to unravel the changes in PIC occupancy when TEC escape is affected. The changes in TEC escape — in particular Elf1 EI — were positively correlated with changes in the transcriptome between the two conditions (Fig. 5b) to a similar extent as were changes in TFB occupancy. This suggests that TEC escape is an integral part of the transcriptional stress response.

A reduction in TEC escape (RNAP and Spt4/5 EI, but not Elf1 EI) was generally associated with the accumulation of TFB and TFEβ at the promoter (Fig. 5b). The *rrn* promoter is one of the strongest promoters in *Saccharolobus*. RNAP and Spt4/5 accumulated at the promoter in response to oxidative stress (RNAP EI from −0.8 to −2.6) suggesting that the control of rRNA synthesis occurs in part at the level of TEC escape (Fig. 5c). TFB accumulated at many promoters showing a strongly reduced TEC escape under oxidative stress conditions such as *gdha-4* and *NuoB* (Fig. 5d, e). In contrast, promoters where TEC escape remained relatively unaffected did not show significant changes in TFB accumulation as in the case of *SSO8549* (Fig. 5f). These promoters were generally characterised by low TEC escape under both growth conditions.

The link between increased TFB accumulation and reduced TEC escape thus offers a possible mechanistic explanation how TEC escape can affect productive transcription. The accumulation of TFB at the promoter in ChIP-seq experiments could reflect a slower progression from the initial formation of ternary DNA-TBP-TFB complexes towards dissociation of TFB from RNAP during promoter clearance. To test whether RNAP recruitment to DNA-TBP-TFB is impaired, we compared TFB

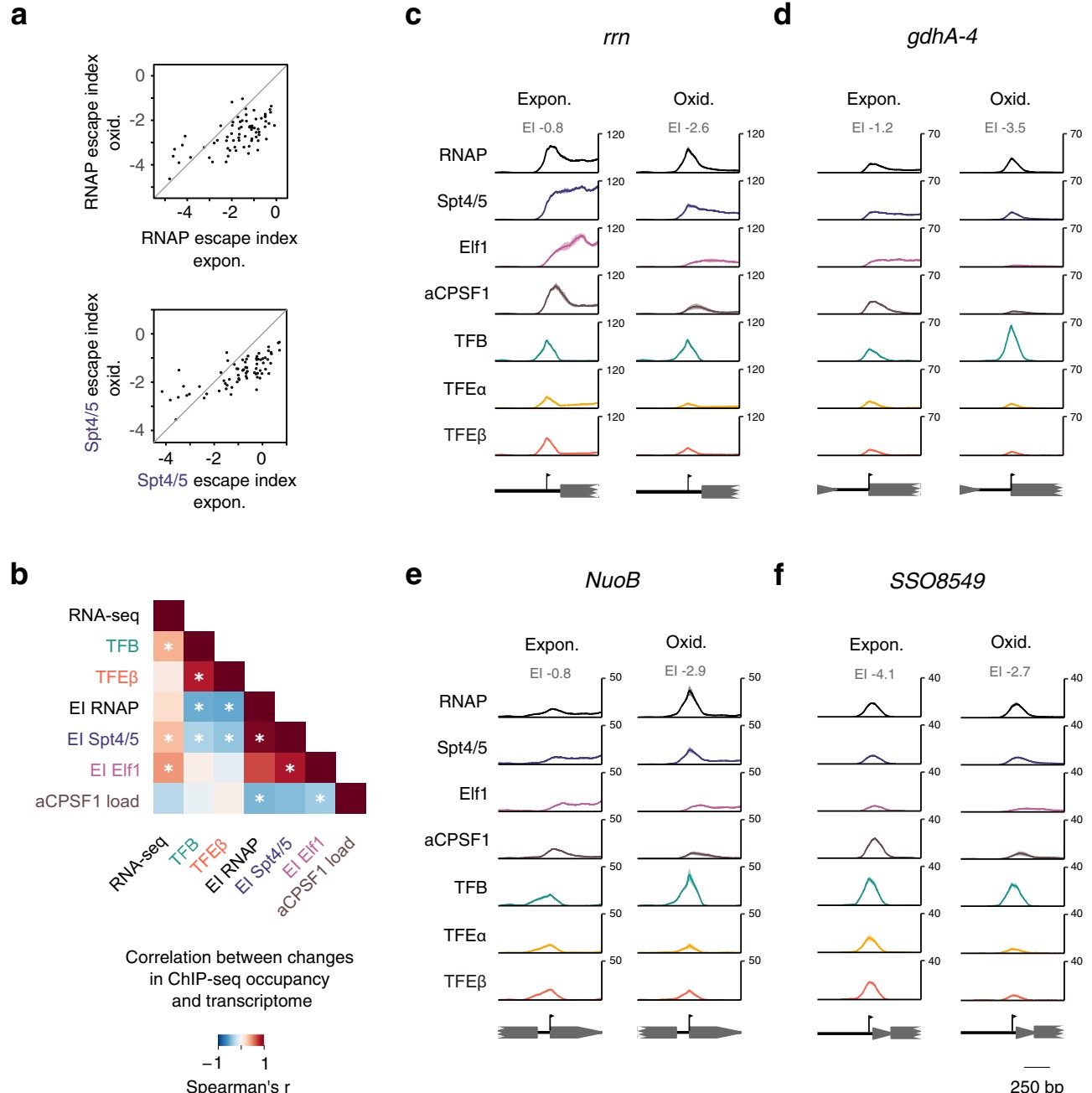

**Fig. 5 TEC escape changes during oxidative stress response. a** High escape TUs show reduced TEC escape under oxidative stress. Scatter plots comparing escape indices under exponential growth and oxidative stress conditions for 71 TUs accessible for analysis in both conditions. **b** Heatmap showing correlated changes in initiation factor occupancy, escape indices and RNA output between exponential growth and oxidative stress. Spearman rank correlations were calculated for protein-encoding TUs accessible for analysis in both conditions (*n* = 70). Correlations were calculated for the mean escape index and the geometric mean of all other values for two biological replicates. * denotes an adjusted *p*-value < 0.05 after multiple testing correction (Benjamini–Hochberg) and  *p* < 0.05 for two combinations of individual biological replicates tested. **c–f** ChIP-seq profiles of the basal transcription machinery for four different promoters during exponential growth (Expon.), and oxidative stress (Oxid.): *rrn* (**c**), *gdhA-4* (**d**), *NuoB* (**e**), and *SSO8549* (**f**). Traces show mean occupancy for two biological replicates with the range depicted as semi-transparent ribbon.

and RNAP ChIP-exo data regarding the fold-changes in PIC signal between exponential growth and oxidative stress (Supplementary Fig. 13). The TFB and RNAP ChIP-exo signals showed equal changes between the two growth conditions independent of the co-occurring changes in TEC escape, which does not support the notion that lowered TEC escape is associated with impaired RNAP recruitment to DNA-TBP-TFB complexes. Alternatively, reduced TEC escape could block PICs during initial transcription or promoter clearance. The finding that PICs accumulate to

higher levels when TEC escape is reduced indicates that changes in RNAP escape indices (which integrate both PIC and promoter-proximal TEC signal) reflect both cause and effect of TEC escape regulation. Curiously, changes in RNAP EIs do show a weaker correlation to transcriptome changes compared to Spt4/5 and Elf1.

In contrast to TFB, changes in TFEβ occupancy did not show any significant correlation to transcriptome changes. We reasoned that the observed heterogeneity in TFEβ promoter

## a

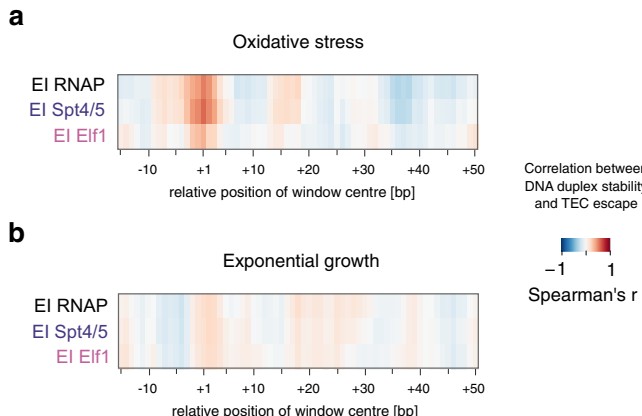

Oxidative stress

EI RNAP
EI Spt4/5
EI Elf1

-10    +1   +10   +20   +30   +40   +50
relative position of window centre [bp]

Correlation between
DNA duplex stability
and TEC escape

−1        1
Spearman's r

## b

Exponential growth

EI RNAP
EI Spt4/5
EI Elf1

-10    +1   +10   +20   +30   +40   +50
relative position of window centre [bp]

**Fig. 6 DNA duplex stability around the TSS is linked to TEC escape. a**, **b** TEC escape is sensitive to DNA duplex stability around the TSS under oxidative stress. DNA duplex stability was calculated over a 7-bp sliding window for individual promoters and correlated with the escape indices for RNAP, Spt4/5 and Elf1 (mean of two biological replicates) under oxidative stress conditions (**a**) and during exponential growth (**b**). Selected TUs with mapped TSS were included ($n = 93$ for oxidative stress and $n = 140$ for exponential growth).

occupancy relative to TFB might be rather the result of TFEβ loss in stalled PICs resulting from low TEC escape, rather than promoters showing different affinities for TFEβ under oxidative stress. This does not preclude a broader, genome-wide effect of TFE depletion on transcription initiation during oxidative stress.

In summary, our results suggest that TEC escape is modulated during oxidative stress, and that environmental insults lower the TEC escape and concomitant RNA output in archaea.

**Stability of the upstream DNA duplex affects TEC escape.** Ultimately, the differences in TEC escape are directly or indirectly dictated by the promoter and template sequence context. This includes the promoter-proximal accumulation of PICs and TECs observed under oxidative stress (Fig. 5a, b). Neither the strength of the BRE-TATA promoter element nor its spacing relative to the TSS showed any significant correlation to any feature of TEC escape (Supplementary Fig. 14). In contrast, we found that oxidative stress-specific accumulation of RNAP and Spt4/5 in the promoter region appeared to be influenced by the stability of the DNA duplex across the TSS. To this end, DNA duplex stability of individual promoters was calculated as the inverse of the predicted Gibbs free energy for a 7-bp sliding window within the promoter-proximal region. Under oxidative stress conditions, DNA duplex stability across the TSS showed a robust correlation with RNAP and Spt4/5 escape indices, but much less so with Elf1 (Fig. 6a) (maximum spearman's $r = 0.51$, $r = 0.57$ and $r = 0.35$, respectively). Notably, under exponential growth conditions, TSS duplex stability showed a weaker, non-significant positive correlation with TEC escape (Fig. 6b) suggesting that the early steps of TEC assembly become sensitive under oxidative stress conditions resulting in the accumulation of TEC$_{Spt4/5}$ and PICs. Consistent with that notion, TSS DNA duplex stability is directly correlated with the ratio of Elf1 to Spt4/5 promoter occupancy under oxidative stress conditions (Supplementary Fig. 15). These results indicate that early transcription elongation and TEC assembly are enhanced by the stable reannealing of upstream DNA but only during the altered conditions of oxidative stress when TFE α/β appears to be depleted from PICs.

**Multiple regression analysis reveals changes in PIC composition with low TEC escape.** Two features are interfering with a quantitative analysis of the relationships between different components of the basal transcription machinery: the non-normal distribution of ChIP-seq occupancy data and the widespread collinearity between occupancy data for different factors. To provide a more comprehensive view of the changes at promoters with high or low TEC escape, we performed a multiple regression analysis for TEC escape (represented by Spt4/5 occupancy data, see methods) under exponential growth and oxidative stress conditions using negative binomial generalised linear models. The models reproduced the observed accumulation of TFB when TEC escape is low (indicated by the negative coefficient for TFB in the model Supplementary Fig. 16). Furthermore, the models reproduced the growth condition-specific effects of aCPSF1 load (as ratio aCPSF1$_{Pr}$ to Spt4/5$_{Pr}$) and DNA duplex stability around the TSS. An increased aCPSF1 load was associated with lower TEC escape specifically under exponential growth conditions. TSS DNA duplex stability was associated with increased TEC escape specifically under oxidative stress conditions (Supplementary Fig. 16).

The model for oxidative stress revealed new insight into the PIC composition associated with TEC escape. TFEβ accumulates at the promoter similar to TFB when TEC escape is low (Fig. 5b). The model suggests that TUs with low TEC escape do have a lower fraction of PICs containing TFEβ. The causative relationship of this relative change in PIC composition could work in either direction. Slow TEC escape could impair PICs assembled on the promoter from completing initiation. This retention could lead to the loss of TFE similar to stalled open PICs of yeast RNAPII that have been shown to lose TFIIE in vitro[62]. Alternatively, low TFE occupancy and slow TEC escape could both be a result of slower transcription initiation. In summary, the multiple regression analysis proposes a link between PIC composition and TEC escape.

## Discussion

**Transcription regulation in archaea.** Transcription in all domains of life has to be fine-tuned over a wide range of synthesis rates that can respond to environmental cues. Compared to Bacteria as well as Eukaryotes (RNAP II), archaeal promoters show a lower apparent complexity in terms of promoter element composition[35,36]. What we know thus far is that archaeal transcription is regulated via enhancing or impairing the recruitment of PICs to the promoter[39–42]. To date, there are no known examples of transcription attenuation or antitermination mechanisms in archaea. How does the archaeal transcription machinery generate the diversity in promoter strength and regulation? Our results point to promoter-proximal elongation as key target in determining promoter strength (Fig. 7). DNA duplex stability around the TSS is likely to constitute a conditional promoter element that plays an important role during oxidative stress where it affects the first stages of TEC escape. The underlying mechanism remains to be explored yet but is likely based on the reannealing of upstream DNA within the Inr region during promoter clearance towards early TEC progression. This is consistent with data from *E. coli*, where duplex stability of the initial transcribed region affects promoter escape in vitro[63]. How does oxidative stress cause promoter-proximal TECs to become sensitive to upstream DNA duplex stability? Changes in DNA topology as observed in *E. coli*[64], ribonucleotide concentrations and DNA chromatinization are possibly contributing factors. However, our transcriptome data do not support strong induction of topoisomerase expression upon oxidative stress.

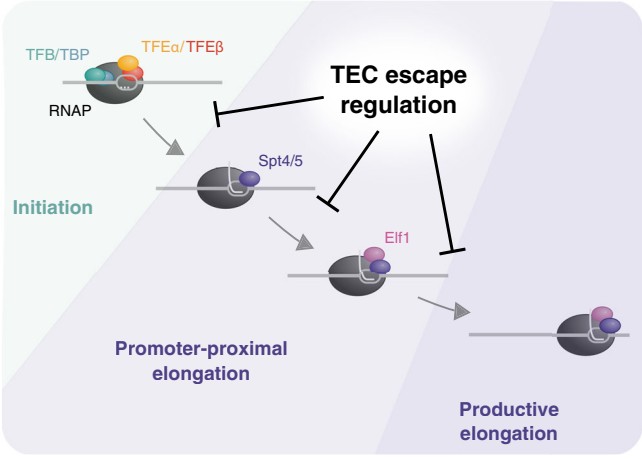

**Fig. 7 A model for the promoter-proximal elongation phase.** Schematic overview of the promoter-proximal elongation phase and the effect of TEC escape regulation on individual steps leading towards productive transcription. Low TEC escape is associated with the accumulation of PICs and the two different TEC intermediates TEC$_{Spt4/5}$ and TEC$_{Spt4/5-Elf1}$.

**Mechanisms underlying promoter-proximal TEC dynamics.** Besides premature termination, the promoter-proximal accumulation of TECs can be explained by slow elongation or pausing that might result in backtracking. Bacterial cleavage factor GreB facilitates the release of *E. coli* RNAP from Sigma70-dependent pause sites[65]. Likewise, cleavage factor TFIIS facilitates the release of promoter-proximally paused RNAP II in *Drosophila* and human cell lines[66,67]. It is tempting to speculate that TFS, the archaeal TFIIS homologue, might play a role in controlling promoter-proximal TEC dynamics in *Saccharolobus*. Unfortunately, our TFS ChIP-experiments were not successful and TFS association with promoter-proximal TECs remains to be tested.

**Functional interactions between initiation and elongation complexes.** Promoter-proximal enrichment of TECs can be caused by altered dynamics such as slower elongation rates or pausing, or premature termination[1]. Importantly, altered TEC dynamics will only affect productive transcription if it leads to a reduced initiation frequency as in the case of promoter-proximally pause RNAPII that blocks PIC formation in metazoans[4,5]. In *S. solfataricus*, promoter-proximal TEC accumulation coincides with accumulation rather than depletion of PICs. This indicates functional interaction between PICs and TECs and we propose that promoter-proximal TECs might prevent PICs from completing initiation and clear the promoter. Notably, our ChIP-exo data reveal broader PIC footprints than previously anticipated based on in vitro data. This could be due to wrapping of the downstream DNA around the PIC[68] or additional DNA-binding factors in the cross-linked complexes such as chromatin proteins[69]. But, independent of the mechanism, it indicates a larger spatial overlap and thereby interference between PICs and promoter-proximal TECs possibly forming the basis for their functional interaction.

**The conserved transcription elongation factor Elf1.** The function of Elf1 remains poorly understood and our data provide the first insight into archaeal Elf1. Yeast Elf1 is recruited after Spt4/5 to the TEC with a gradual increase in Elf1 occupancy towards the poly-adenylation site[70] and Elf1 recruitment depends on Spt4[71]. In *S. solfataricus*, Elf1 is recruited subsequent to Spt4/5 in the promoter-proximal region. The role of Elf1 as an integral part of the TEC genome-wide makes it a likely target for regulation,

possibly by post-translational modification of the N- and C-terminal tails of Elf1[72].

**Evolution of promoter-proximal regulation in archaea and eukaryotes.** The pivotal role of Spt4/5, Elf1 and aCPSF1 in the early TEC dynamics in *Saccharolobus* shows intriguing parallels to metazoans. Firstly, the early elongation phase of transcription is rate limiting for gene expression. Secondly, Spt4/5 (DSIF) is an integral component of promoter-proximal elongation complexes in humans as well as archaea. Thirdly, CPFS73-related RNases are likely to mediate premature termination of promoter-proximal RNAPs in both eukaryotes[10,73] and archaea. Our discovery that early elongation complex dynamics modulate transcription in *Saccharolobus* suggests that promoter-proximal regulation is an ancient feature of the archaeo-eukaryotic transcription machinery. Control of promoter-proximal TEC escape efficiency could provide a simple primordial mechanism for gene regulation, from which a more complex process evolved that involves the stable pausing of elongation complexes, and a tightly controlled pause-release by factors including NELF and P-TEFb.

In Sum, we provide evidence for widespread promoter-proximal transcription regulation in archaea. Our data suggest that the archaeal transcription cycle involves at least two major regulatory checkpoints: (i) recruitment of RNAP and initiation factors to the promoter and (ii) TEC escape into productive elongation likely involving a negative feedback effect on initiating RNAPs upstream as well as premature termination. Together they create a dynamic mosaic of mechanisms that determines the transcription output in archaea. The relative simplicity and biochemical tractability of archaeal transcription complexes provides for the development of in vitro models to elucidate on the molecular mechanisms underlying TEC escape.

## Methods

**ChIP-seq.** Rabbit antisera against *S. solfataricus* TFB, TBP, TFEβ, TFEα and Rpo4/7 have been described previously[24]. Polyclonal rabbit antisera against recombinant Spt5, Elf1, and aCPSF1 were produced at Davids Biotechnology (Regensburg, GER). All antibodies were purified from antiserum by Protein A-agarose affinity chromatography.

*S. solfataricus* P2 cells were grown in Brock medium[74] at 76 °C in a Thermotron air incubator (Infors) to mid-exponential growth phase (O.D.$_{600}$ 021–0.29). For oxidative stress, cells were grown overnight in modified Brock medium without FeCl$_2$ supplemented with 0.2% tryptone to mid-exponential growth phase (O.D.$_{600}$ 0.11 to 0.24) before the addition of 30 μM H$_2$O$_2$ similar to what has been described before[75]. 105 min after H$_2$O$_2$ addition cells were cross-linked. All cultures were cross-linked by the addition of 0.4% formaldehyde for 1 min before quenching with 100 mM Tris/HCl pH 8.0.

Cells were washed three times in PBS buffer before freezing in liquid nitrogen and storage at −80 °C. To prepare lysates for ChIP experiments, cells were resuspended in lysis buffer (50 mM HEPES/NaOH pH 7.5, 140 mM NaCl, 1 mM EDTA, 0.1% Na-deoxycholate and 1% Triton X-100) supplemented with complete protease inhibitor cocktail (Roche). DNA was sheared in polystyrene tubes in a Q700 cup sonicator (Qsonica) at 4 °C to an average fragment size of 150 bp as judged by agarose gel electrophoresis. Debris was removed by centrifugation before freezing in liquid nitrogen and storage at −80 °C. For ChIP, 500 μl lysate diluted to a DNA content of 20 ng/μl (based on A260 measurements) was supplemented with antibody (2 μg of TFB or Rpo4/7 antibodies, 4 μg for TFEα or TFEβ) and incubated overnight in an end-over-end rotator at 4 °C. After addition of 50 μl of sheep anti-rabbit IgG Dynabeads M-280 (Thermo Scientific) were added and incubation was continued for 1 h. Beads were washed twice with 1 ml lysis buffer, once with lysis buffer containing 500 mM NaCl, wash buffer (10 mM Tris/HCl pH 8.0, 100 mM LiCl, 1 mM EDTA, 0.5% Na-deoxycholate and 0.5% Nonidet P-40) and TE buffer. Immunoprecipitated material was eluted from beads by the addition of 200 μl ChIP-elution buffer (50 mM Tris/HCl pH 8.0, 10 mM EDTA and 1% SDS), de-cross-linked overnight at 65° in the presence of 10 μg RNase A and 40 μg Proteinase K. DNA was purified using Qiaquick PCR purification kit (Qiagen). For Spt4/5, Elf1 and CPSF1 ChIP experiments, the lysate volume was increased to 1 ml and 8 μg of antibody were used in combination with Protein G Dynabeads M-280 (Thermo Scientific) as described above. The yield of the ChIP experiments was determined using Qubit dsDNA HS (Thermo Scientific). Libraries were prepared using the NEBNext Ultra II DNA library prep kit for Illumina (NEB) according to the manufacturer's protocol. Library quality and quantity was assessed using

Agilent High Sensitivity DNA kit (Agilent Technologies) and Qubit dsDNA HS assay kit (Thermo Scientific).

**ChIP-seq read mapping and fragment size distribution adjustment**. We generated paired-end ChIP-seq data on Illumina HiSeq platforms with two biological replicates per condition. 75 or 125 nt reads were trimmed from the 3′-ends to 50 nt read length and the read pairs were aligned to the *S. solfataricus* P2 genome (NC_002754.1) using bowtie[76] (parameters -v 2 -m 1–fr) retaining only alignments for read pairs with a single best match. The alignments were converted to bam file format using SAMtools[77]. To ensure comparability between different ChIP-seq samples, we sequenced all samples to high genomic coverage and adjusted the fragment size distribution using a computational approach by subsampling the reads. The fraction size range in a ChIP-seq experiment is shaped in different ways including the chromatin shearing, size selection method and conditions during the library preparation, and gating (minimal and maximal fragment sizes) during read alignment and calculation of genomic coverage.

In order to adjust the fragment size distribution computationally, bam files were converted to bedpe format using BEDTools bamtobed[78] and imported into R. We adjusted the fragment size distribution to fit a normal distribution with mean 120 and standard deviation of 18. To this end, then read pairs were binned into 200 bins according to fragment size in the range of 51–250 bp randomly drawn from each bin using the sample() function in R. The number of reads to be drawn was determined by their relative frequency in the target distribution (dnorm()) multiplied by maximum total number of reads possible without exhausting any of the bins. This procedure retained on average 46% of the read pairs. Data were exported as bed files with the fragment coordinates and subsequently converted back to bam format for any downstream analysis.

In order to assess the reproducibility of the data across biological replicates, we calculated read coverage using DeepTools bamCoverage and multiBigWigSummary[79] with bin size 50. Data were imported into R and pairwise correlation between unfiltered and sampled data as well as the pairwise correlation between biological replicates was assessed using the cor.test() function (see Supplementary Figs. 17 and 18).

**Peak calling**. Peaks were identified with MACS2[80] in BEDPE mode, $q$ 0.01 and with the call-summit sub-function in order to identify overlapping peaks. MACS2 output provides summit coordinates and quality scores for each peak, but the coordinates for each enriched region are not split between the overlapping peaks. For this reason, we used the peak summit positions to merge peaks from replicates with 40 bp max distance which should correspond to more than 50% overlap between the peaks using BEDTools window function[78]. For the reproducibility analysis of the peaks between replicates based on p-values[81], we set a global IDR threshold of 0.05 using the Cran IDR package in R[82]. A number of reproducible TFEα and TFEβ peaks within the rRNA operon were removed as this region exhibited an overall strongly increased background. Finally, for reproducible peaks the average position and fold-enrichment between the replicates was calculated. All spearman pairwise correlations were calculated using the spearman.ci() function from the RVAideMemoire package in R with confidence intervals calculated using bootstrapping ($n = 1000$). Spearman correlation estimates were considered to be significantly different at significance level α when the confidence intervals calculated from the bootstrapped data set for the same significance level were non-overlapping for both biological replicates.

**Occupancy data plotting**. Bam files were normalised against input using deep-Tools bamCompare using the SES method for scaling (10,000 bins, 200 bp bin width)[79,83] and converted to bigwig format. For individual gene plots normalised bigwig files were imported into R via the rtracklayer package[84] and plotted using ggplot2[85]. Heatmaps were generated using deepTools computeMatrix and plotHeatmap functions[79].

**TU selection and escape index calculation**. Because archaeal genomes are densely packed with TUs, we filtered the set of *S. solfataricus* TUs to ensure robust, TU-specific signal quantification of the ChIP-seq data. We manually curated a previously published map of 2229 TUs based on RNA-seq data[86]. The dataset contained 1035 TUs with primary TSSs. We also included two previously determined transcription start sites for 5S and 16S/23S rRNA genes[87]. For the remaining 1192 TUs no experimentally verified TSS data were available, but based on the generally short 5′ UTR length of Sso mRNAs (69% with length of 4 bp or shorter)[86], we used the start codon position as rough indicator for TSS position with additional adjustments based on our ChIP-exo data (see below). The start codon for genes Sso0845 (TU 565) and Sso1077/*fumC* (TU 710) were reassigned to the third and second ATG codon 33 bp and 21 bp downstream, respectively, as TFB and TFE ChIP-seq, TFB ChIP-exo and permanganate ChIP-seq signals as well as RNA-seq data all consistently suggested the reassignment.

We filtered for those TUs showing TFB occupancy at the promoter with bijective correspondence and pairing of TFB and TFEβ peaks with bijective correspondence. To this end, TFB ChIP-seq peaks were matched to TFEβ peaks and then assigned to TSSs using BEDTools window with the peak summit position within 40 bp maximal distance from the TSS.

In order to ensure reliable data normalisation and quantification over the TU body, we only considered TUs with 500 bp minimum length and a minimum coverage of 20 reads in the chromatin input sample within the −250 to +500 interval relative to the TSS. TUs were further filtered against internal TFB peaks within +40 to 500 bp relative to TSS to ensure that RNAP and Spt4/5 occupancy is not influenced by any TU internal promoters.

Divergent promoters in *S. solfataricus* are often tightly spaced causing the RNAP and Spt4/5 ChIP-seq signal from these promoter pairs to be convoluted to considerable extent. To address this problem, we filtered our TU set further for those where the input-normalised Spt4/5 occupancy at TSS position was at least 1.5x increased compared to occupancy at position −150 relative to TSS.

The resulting set of TUs was checked for consistent positions of the ChIP-exo data as additional control for TSS position (see below).

To determine the escape index, the log2 ratio of input-normalised RNAP occupancy within the TU body (+251 to 500 bp relative to TSS) to the promoter region (−50 to +100) was calculated.

**Positional adjustment of TSS prediction based on ChIP-exo data**. For TSS positions that were initially estimated from start codon positions only, we used TFB ChIP-exo signal on the non-template strand to get a more precise estimate of TSS position. The ChIP-exo TFB signals for genes with experimentally mapped TSSs[86] were used to build a training set. A peak with median position −14 relative to TSS (Inter-Quartile Range −16 to −12) was identified.

For nine TUs out of 82 TUs without experimentally mapped TSS in total, adjustment of TSS positions by 10 to 33 bp were suggested. For TU 1536 no consistent peak could be identified and it was removed from the data set. Lastly, all suggested adjustments of TSS positions were checked manually for their consistence with ChIP-seq, ChIP-exo, permanganate ChIP-seq and RNA-seq data and TSS positions were adjusted further by max 3 bp to match the (C/T)(A/G) consensus of the Inr promoter element.

**RNA 3′-end selection**. To test aCPSF1 association with putative transcription termination sites, we used an initial dataset of 1727 predicted RNA 3′-ends based on the Rockhopper 2[88] output of the RNA-seq data (see below). Predicted RNA 3′-ends were filtered by the following criteria: (i) a TU length of >300 nt, (ii) no TFB peaks within the surrounding 600 bp to filter out aCPSF1 peaks resulting from promoter-proximal recruitment, (iii) continuous input coverage of >20 reads in the surrounding 500 bp to ensure reliable input normalisation, (iv) a two-fold increase in average Spt4/5 occupancy in the 250 bp downstream of the predicted RNA 3′-end compared to the 250 bp window upstream. The final data set comprised 41 predicted RNA 3′-ends.

**TATA-box assignment**. The *S. solfataricus* BRE-TATA box motif was determined by scanning promoters with mapped TSS within a 24-bp window (positions −42 bp to −19 relative to TSS) using MEME[89] in 'oops' mode with 8–15 bp motif width.

**ChIP-exo**. For ChIP-exo analysis, we used the ChIP-exo Kit (Active Motif) according to manufacturer's specifications with the following modifications. This kit is based on the modified ChIP-exo protocol adapted for Illumina sequencing[90]. Cell growth, crosslinking and DNA shearing were carried out as described for ChIP-seq samples, but DNA was sheared to a range of >200 bp to be suitable for ChIP-exo. Immunoprecipitation was carried out as for ChIP-seq samples by incubating 1 ml lysate with 8 µg antibody overnight. The lysates were transferred to a new tube with 50 µl Protein-G Dynabeads (Active Motif) and further incubated for 1 h before following the manufacturer's recommendations for washing of the beads and library preparation. Library quality was assessed using Agilent High Sensitivity DNA it (Agilent Technologies) and Qubit dsDNA HS assay kit (Thermo Scientific). Libraries were sequenced on the Illumina HiSeq platform with 50 cycles read-length. Reads were aligned to the *S. solfataricus* strain P2 genome using bowtie v1.2.2[76] (parameters -v 2 -m 1 –best –strata -S), converted to bam file format SAMtools[77] and converted to strand-specific 5′-end coverage data using MACE v1.2 pre-processor function with default settings normalised to 1x genome coverage[91]. For pooling of replicates we calculated the geometric mean.

**DNA duplex stability**. DNA duplex stability was estimated using a nearest-neighbour model[92]. Gibbs free energy values for each dinucleotide were extrapolated to the optimal growth temperature for *Saccharolobus* of 76 °C. Initially, windows with sizes of 3–20 bp were tested for correlation of duplex stability with escape indices and a 7 bp from position −3 to +4 relative to TSS yielded the highest Spearman 's r. In the nearest-neighbour model, the salt concentration is included in the Gibbs free energy calculation as offset. While the internal salt concentration of *Saccharolobus* cells is not known and could only be roughly estimated, the Spearman correlation is not influenced by offsets.

**Model of productive transcription elongation under oxidative stress**. We generated negative binomial generalised linear models with log link function using

the glm.nb() function in the MASS package in R. Raw Spt4/5 TU body coverage was used as the dependent variable in the models while log-transformed raw input coverage over the same region was included as fixed offset effectively performing the input normalisation. We calculated the total occupancy signal within the TU body (Bd, +250 to +500 bp relative to TSS) for Spt4/5 ChIP-seq data as well as the chromatin input for the subset of TUs with mapped TSSs. These values were corrected for the average fragment length (120 nt) to obtain an estimate of over-lapping read pair counts. Log-transformed input-normalised Spt4/5$_{Pr}$ occupancy was included as a second offset term. Thereby, the models effectively identified variables predictive for TEC escape efficiency. As explanatory variables, we tested TFB and TFEβ promoter occupancy (log-transformed), aCPSF1 load on promoter-proximal TECs (log-transformed aCPSF1$_{Pr}$ to Spt4/5$_{Pr}$ ratio), and TSS DNA duplex stability. The full model was structured as follows:

$$\log(\mathbf{Spt4/5_{Bd}coverage}) \sim \log(\mathbf{input_{Bd}coverage}) + \log(\mathbf{Spt4/5_{Pr}})$$
$$+ \beta_0 + \beta_1 \times \log(\mathbf{TFB})$$
$$+ \beta_2 \times \log(\mathbf{TFE\beta}) + \beta_3 \times \log\left(\frac{\mathbf{aCPSF1_{Pr}}}{\mathbf{Spt4/5_{Pr}}}\right)$$
$$+ \beta_4 \times \mathbf{TSS\ DNA\ duplex\ stability}$$

TUs passing a Cook's distance of 0.5 for a model of all explanatory variables were removed from the dataset. To identify the optimal model, we used the step() function in the MASS package for automatic model building using default settings for AIC minimisation. The null model was used as lower boundary and the full model including all variables as upper boundary. Searches were initiated with the null model (including only the raw input coverage and Spt4/5$_{Pr}$ offsets) as well as all explanatory variables added separately. Statistical significance of the included variables in the model was confirmed using the Likelihood-ratio chi-squared test implemented in anova.negbin() (car package) by comparing the optimal models against models excluding single variables ($p < 0.05$). Only statistically significant explanatory variables consistent between replicates were retained in the final model. To validate the model, bootstrapped 95% confidence intervals were calculated for each coefficient and checked whether they were consistently positive or negative using the Boot() function from the car package.

**Permanganate ChIP-seq.** For permanganate ChIP-seq, we adapted our ChIP-exo protocol analogously to the method described by Gilmour, Pugh and co-workers[52,53] as follows. After crosslinking, cells were washed once in 25 ml ice-cold PBS and resuspended in 25 ml room temperature PBS. 833 µl 300 mM KMnO$_4$ was added to a final concentration of 10 mM and incubated at room temperature. After 1 min, 25 ml of stop solution (PBS supplemented with 0.8 M 2-mercaptoethanol and 40 mM EDTA) was added and cells were washed three more times in PBS before freezing in liquid nitrogen. Cells were further processed as for ChIP-exo, but λ-exonuclease and RecJ$_F$ digestion steps were omitted. After reversal of crosslinks and ethanol precipitation, the purified DNA was dissolved in 10% piperidine and incubated at 90 °C for 30 min. Piperidine was removed by three steps of 1-butanol extraction followed by chloroform extraction before ethanol precipitation. P7 primer extension, P5 adaptor ligation and PCR amplification were carried out according to the ChIP-exo protocol. Reads were mapped onto the S. solfataricus strain P2 genome as described above and converted to strand-specific 5′-end coverage data using BEDTools genomecov. Coverage data were loaded into R and corrected for the position by −1 nt so that it corresponds to the modified nucleotides eliminated during permanganate/piperidine cleavage. Coverage data were filtered for genomic positions with Ts. While regions around TSSs are generally characterised by strongly increased signal on T positions over non-T positions (expected for potassium permanganate/piperidine treatment of DNA), we found that these regions were often flanked by broader regions of lower coverage signal that did not show any specificity for Ts and possibly results from incomplete P7 adaptor ligation during the library preparation. For this reason, we performed a background correction on the data. Background signal was calculated as from four neighbouring non-T positions for each T (the two closest non-T residues on either side). The median background signal was subtracted and the resulting T-specific signal was normalised to 1x genome coverage of Ts for both strands.

**RNA-seq.** RNA was isolated by mixing samples directly with three volumes ice-cold TRIzol LS (Thermo-Fisher). RNA was isolated according to manufacturer's protocol and remaining genomic DNA was removed using the TURBO DNA-free kit (Thermo-Fisher). RNA was quantified using the Qubit RNA BR Assay kit (Thermo Scientific) and quality was assessed using the RNA ScreenTape system (Agilent). Libraries were prepared at Edinburgh Genomics using the TruSeq® Stranded Total RNA Library Prep kit (Illumina) including partial ribosomal RNA depletion with the Ribo-Zero rRNA Removal Kit (Bacteria) (Illumina). 75 bp paired-end reads were generated on a HiSeq 4000 system (Illumina). Coverage tracks were produced using the Rockhopper 2 software package in–rf mode[88]. Rockhopper 2 quantifies transcripts by applying an upper quartile normalisation. The coverage tracks with raw coverage were corrected for sequencing depth and the fraction of reads mapping sense to protein encoding genes (mRNA) yielding normalised coverage in counts per million (cpm). Reproducibility between

biological replicates was assessed based on the correlation of raw count data for 3057 coding genes (see Supplementary Fig. 19).

**Cappable-seq short RNA sequencing.** 15 ml cell culture was rapidly mixed with 30 ml pre-cooled RNAprotect Bacteria Reagent (Qiagen) placed in an ice bath. Cells were harvested by centrifugation (5 min at 4000 × g at 4 °C). Pellets were immediately subjected to RNA isolation using the mirVana miRNA isolation kit (Ambion) with an initial resuspension buffer volume of 200 µl following the protocol for small RNAs (20-200 nt length). Library preparation and deep sequencing was conducted at Vertis Biotechnologie (Germany). In brief, 5′-triphosporylated RNA was capped with 3′-desthiobiotin-TEG-GTP (NEB)) using the Vaccinia virus Capping enzyme (NEB) and biotinylated RNA species were subsequently enriched by affinity purification using streptavidin beads yielding 0.6–1.3% of the sRNA preparation. The eluted RNA was poly-adenylated using E. coli Poly(A) polymerase and 5′-ends were converted to mono-phosphates by incubation with RNA 5′ Pyrophosphohydrolase (NEB). Subsequently, an RNA adaptor (5′-ACACTCTTTCCCTACACGACGCTCTTCCGATCT-3′) was ligated to the newly formed 5′-monophosphate structures. First-strand cDNA synthesis was performed using an oligo(dT)-adaptor primer and the M-MLV reverse transcriptase at 42 °C. The resulting cDNA was finally PCR-amplified (12 cycles) with TruSeq Dual Index sequencing primers (Illumina) and Herculase II Fusion DNA Polymerase (Agilent). The libraries were sequenced on an Illumina NextSeq 500 system with 75 bp read length. In order to remove poly(A)-tails and adaptors, we trimmed the reads using Cutadapt[93] in two rounds with the following settings to prevent trimming of naturally occurring A-rich RNAs due to the low GC-content of the S. solfataricus genome: (i) -a "{A15}" -e 0 -m 15 to remove all poly(A) stretches of at least 15 nt length plus downstream regions and (ii) -a "A{15}"X -e 0 -O 5 to terminal shorter poly(A) stretches of minimum 5 nt length. Trimmed and untrimmed reads were split into separate fastq files using awk. Both fastq files were aligned to the S. solfataricus genome using bowtie v1.2.2[76] (parameters -v 1 -m 1 --best --strata -S) with untrimmed 75 nt reads shortened to 71 nt (−3 4). The bam file output was merged, sorted and indexed using SAMtools[77]. Bam files were imported into the R environment using the rsamtools and GenomicRanges packages and filtered to ensure a unique sequence of the initial 20 bp within the S. solfataricus genome required to map the reads in R using the Biostrings package. TSS-RNA were defined as RNAs with a 5′-end within 20 nt of a mapped or predicted TSS. The two biological replicates showed good reproducibility of TSS-RNA occupancy with a Spearman correlation of 0.98 for 438 mappable promoters.

To calculate the fraction of TSS-RNAs with a length shorter than 50 nt for each TU, a minimum read count of 10 TSS-RNAs per TU per replicate was used and values were averaged between the two biological replicates.

**Immunodetection.** Cell lysates were resolved on 12% Tris-tricine SDS gels and blotted onto nitrocellulose membranes. All immuno-detections were carried out using polyclonal antisera (see above) in combination with donkey anti-rabbit IgG Dylight680 (Bethyl Laboratories). Dps-l antiserum was a kind gift of Mark Young (Montana State University, USA). As loading control, we used sheep Alba anti-serum (kind gift of Malcolm White, University of St. Andrews, UK) in combination with donkey anti-sheep IgG Alexa488 (Thermo Fisher). Blots were scanned on a Typhoon FLA 9500 scanner (GE Lifesciences).

**In vitro transcription.** In order to trace promoter-proximal transcription elongation dynamics in vitro, we developed a cell lysate-based synchronised transcription assay. Synchronisation was achieved by supplementing an inactive variant of initiation factor TFB comprising only the C-terminal cyclin folds (TFBc) simultaneously with rNTPs. TFBc forms ternary complexes with TBP and DNA containing TATA/BRE promoter motifs, but it fails to recruit RNA polymerase and does not facilitate transcription initiation[60,94].

To generate cell lysates for the in vitro transcription, 4 litre S. solfataricus P2 cultures in exponential growth phase were transferred to an ice-water bath for cooling. Cells were harvested by centrifugation for 20 min at 4000 × g at 4 °C and washed three times in ice-cold 20 mM sucrose solution before flash freezing in liquid nitrogen and storage at −80 °C. Cells were resuspended in 6 ml 10 mM MOPS pH 6.5, 10 mM MgCl$_2$, 1 mM DTT, 1 mM EDTA, 0.1% Triton X-100 and incubated for 1 h on ice for cell lysis. Insoluble material was removed by centrifugation for 20 min at 20,000 × g at 4 °C and the supernatant was aliquoted and stored at −80 °C.

Templates for in vitro transcription were generated by PCR-amplifying promoter regions −50 to +100 relative to the TSS for thsB, rps8e, dhg-1 and CRISPR C from S. solfataricus P2 genomic DNA and ligating the products into the vector pGEM-T (Promega) (see Supplementary Table 1 for details).

Before using the cell lysates in in vitro transcription reactions, nucleotides were degraded by treating the lysate with 100 unit/ml recombinant shrimp alkaline phosphatase (New England Biolabs) for 20 min at 37 °C followed by 5 min at 65 °C to inactivate the phosphatase. Samples contained 2/3 volume cell lysate, 20 ng/µl linearised plasmids, 2 mM spermidine, 10 mM MOPS pH 6.5 and 10 mM MgCl$_2$. For synchronised transcription assays, samples were incubated for 1 min at 65 °C to assemble pre-initiation complexes. 50 µM rNTPs including trace amounts

[α-$^{32}$P]-UTP (Perkin Elmer) were provided to initiate transcription together with 5 μM TFBc blocking further pre-initiation complex assembly. At given time points, 30 μl sample were withdrawn and mixed rapidly with 200 μl stop mix (20 mM EDTA pH 8.0, 200 mM NaCl, 1% SDS, 250 ng/μl torula yeast RNA and 0.1 mg/ml Proteinase K). After 5 min incubation, samples were purified by a single acidic phenol/chloroform extraction step. Transcripts were subsequently enriched by affinity purification following a protocol based on[95]. 200 μl supernatant from the phenol extraction step was mixed with 100 μl salt adjust mix (30 mM Tris/HCl pH 8.0, 500 mM NaCl) and 1.6 pmol 3′-biotinlylated antisense oligonucleotide matching the first 25 nt of the transcripts (based on the predicted TSS) (Supplementary Table 1). 3′-biotinylation of the oligonucleotides was carried out with biotin-14-dATP (Jena Bioscience) and Terminal transferase (New England Biolabs). Transcripts and biotinylated oligonucleotides were allowed to hybridise overnight at room temperature. Samples were mixed with 120 μl of 1.25 mg/ml Dynabeads MyOne Streptavidin C1 (ThermoFisher) pre-equilibrated in 2x B&W buffer (10 mM Tris/HCl pH 8.0, 1 mM EDTA, 2 M NaCl). Beads were washed twice with 300 μl washing buffer (10 mM Tris/HCl pH 8.0, 5 mM EDTA, 10 mM NaCl, 100 ng/μl torula yeast RNA) and finally eluted in 15 μl formamide sample buffer (95% deionised formamide, 18 M EDTA, 0.025% SDS) for 5 min at 95 °C. 10 μl of the samples were resolved on 12% polyacrylamide, 7 M Urea, 1× TBE sequencing gel. Transcripts were detected by phosphor imagery and quantification of bands was performed using the ImageQuant TL software (GE Life Sciences).

**Reporting summary**. Further information on research design is available in the Nature Research Reporting Summary linked to this article.

## Data availability
The data that support this study are available from the corresponding author upon reasonable request. All sequencing data files (ChIP-seq, ChIP-exo, permanganate ChIP-seq, RNA-seq and Cappable-seq) and the processed data were deposited at NCBI GEO under accession code GSE141290.

## Code availability
The analysis code is available on Zenodo[96] [https://doi.org/10.5281/zenodo.5196117].

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

## Acknowledgements

We thank Declan Barker for technical assistance. We also would like to thank Amy Schmid (Duke University), Mark Williams (Birkbeck College) for their advice. We also thank Jonathan Chubb, Jürg Bähler (UCL), Alan Cheung (University of Bristol) and Phil Robinson (Birkbeck College) for critical reading of the manuscript. Research in the RNAP laboratory at UCL is funded by a Wellcome Investigator Award in Science 'Mechanisms and Regulation of RNAP transcription' to FW (WT 207446/Z/17/Z).

## Author contributions

F.B. and F.W. conceived the study; F.B., T.F. and K.S. performed experiments; F.B. and D.M. analysed the data; F.B. and F.W. wrote the manuscript.

## Competing interests

The authors declare no competing interests.
