## [Peer Review File · Nature Communications]

Promoter-proximal elongation regulates transcription in archaeaReviewers' Comments:

Reviewer #1:

Remarks to the Author:

The authors describe an accumulation of promoter proximal transcription elongation complexes (TEC) in the archeon *Sulfolobus solfataricus*. Based on a number of high throughput sequencing experiments they argue that TEC escape into productive elongation is a rate limiting step in transcription that provides an additional level of regulation to archaean transcription. They show the recruitment of transcription elongation factors Spt4/5 and Elf1 along with termination factor aCSPF1 to the early TEC and suggest that the regulation might occur via blocking upstream initiating RNA polymerase (RNAP) or premature transcription termination.

This work provides a wealth of high throughput data and in general makes a compelling case for the functional importance of promoter proximal pausing in archaea. However, there is little explanation presented with respect to the mechanism of the demonstrated TEC accumulation. The Methods section describing the computational work could benefit from additional clarification, perhaps in the form of a supplemental text. This is especially true for the extensive raw data preprocessing steps that sometimes result in the discard of more than half of the reads (in particular, when adjusting the fragment size distribution). The inclusion of a github repository with some of the code used to perform analysis and generate figures is a welcome addition, but that too should be improved by additional documentation.

The authors first use the ChIP-exo signal from the basal initiation factors TFB and TFE as well as from RNA polymerase to examine the footprint of the promoter initiation complex (PIC) around TSS and then invoke the ChIP-Seq data for RNAP and transcription elongation factors Spt4/5 and Elf1. The resulting profiles seem to suggest the accumulation of RNAP along with these two factors proximal to the promoter, but well outside of the PIC complex. In addition, there is a signal for the termination factor aCSPF1 with its peak corresponding to the apparent peak of RNAP. This suggests a possible interplay between premature termination and RNAP pausing. Does RNAP backtracking play a role? Some discussion and relevant references would be appropriate. The addition of native elongating transcript sequencing (NET-Seq) data would help to elucidate both the functional status of RNAP and the underlying mechanism.

The authors further split transcriptional units into three groups with "low", "intermediate" and "high" efficiency of TEC escape into productive elongation and quantify this by computing EI (Escape Index) as a log-transformed ratio of average RNA polymerase occupancy in TU body over the promoter region. The attempt to show the correlation between EI and the fraction of "short" RNAs synthesized as assayed by TSS-RNA seq is not very convincing. The statistical significance is apparent but the size of the effect is modest and the biological relevance is questionable. This is further confirmed by the relatively low correlation between RNAP and Spt4/5 EI and RNA output in the comparison of these values during oxidative stress shown in Figure 5.

The purpose of the proposed multiple regression statistical model is not clearly explained. Was it meant as a predictive model? How was it trained and how was its performance validated? Was it meant to select the variables that explain most of the variance in the data?

Minor:

Line 448: "We generated paired-end HiSeq data from our ChIP-Seq data..." - this obviously a typographical error.

Line 449: "Reads were trimmed to 50 nt..." - trimmed how? On the 5' ends? What was the original read length?

Line 500: what is "filtering for consistency"?

Figure 3 ab and Figure 4: The description is very confusing. If the data shown are from a single biological replicate then what are the mean and standard deviation?

Reviewer #2:

Remarks to the Author:

Archaea constitute one of three domains of life, and their basal transcription system has been characterized as a hybrid of eukaryotic and bacterial transcription systems. Study and understanding of transcription regulation mechanisms in archaea is important to highlight the basic transcription regulation mechanisms across all three domains of life, and give insights into functional evolution of multi-subunit polymerases transcription machinery.

The present study "Promoter-proximal elongation regulates transcription in archaea" by Blombach et al. focuses on how the transcription machinery is distributed along genes in the crenarchaeon *Sulfolobus solfataricus*. The authors utilized functional genomics approaches to reveal how transcription elongation complex (TEC) escape is rate-limiting for transcription read-out during exponential growth of *S. solfataricus*. By using ChIP-seq and ChIP-exo experiments, the authors showed a discrete footprint of RNA polymerase and different transcription initiation factors around the transcription start site (TSS), and their uniform assembly at promoter loci. They also reveal that the transition of RNAP by promoter escape into productive and processive elongation differs greatly across different transcriptional units. The results described in the manuscript establish that TEC escape is an important factor for determining promoter strength and transcript levels. The authors further compared the TEC escape during exponential growth and under oxidative stress conditions, surmising that the TEC escape is an integral part of the transcriptional stress response.

Overall, this is an extensive study by genomics approach investigating regulatory checkpoints in archaeal transcription and provides the basis for further in-vitro studies to decipher the underlying molecular mechanisms of TEC escape. Following points need to be addressed before considering for publication:

1. Line 26: Rephrase: "Transcription output is determined by the frequency of transcription initiation and premature termination". It is unclear if "output" here is the net turnover or the overall rate.
2. Authors showed RNAP is not effective in escaping into productive transcription in some TUs. But authors did not really explain why. What are the possible reasons? Do the authors think these results stay the same throughout different growth conditions?
3. Line 255-258: The results show that RNAP TEC escape is reduced in oxidative stress. But what could be the reason? Reason for accumulation of RNAP and Spt4/5 at promoters in oxidative stress conditions? How does an oxidative environment affect the canonical transcription pathway? Is this change could be archaea specific or omnipresent?
4. Line 264: "The accumulation of TFB and other initiation factors": this sentence is not clear. Do authors mean that more than one TFB/other factors recruited to PIC and form non-functional PIC? Or a PIC is stuck at transcription initiation stage? Authors should shed light on what are the key known changes under oxidative stress conditions. For e.g., NTP concentrations/less protein synthesis etc.?
5. Line 345: The authors suggest DNA duplex stability controls TEC escape. Can authors show the sequence comparison of some example promoters? How much difference in the stability or what percentage of G-C and A-T contents is considered as stable and unstable?

6. Through proofreading of manuscript is required for typos and article.

7. *Sulfolobus solfataricus* has been called as *Saccharolobus solfataricus* since 2018. Authors doesn't have to use this new name but better to introduce it at the beginning of the manuscript.

8. In the beginning of this manuscript, authors used a term "TEC escape" but it is not commonly used in transcription research field. Usually called as "promoter escape" and "RNAP escape" as authors used later in this manuscript. Authors may have a reason to call this process as "TEC escape". If so, define TEC escape.

Minor comments:

9. Line 29: metazoans

10. Line 90, 98 Figure 1B.....1b

11. Line 114: Thymine basis..... bases

12. Line 120 – 122: The permanganate footprinting data is from this manuscript. "Figure 1F" citation is missing.

13. Line 192-194: references are missing

14. Figure 2h r 014..... 0.14

15. Supplementary figure S7: 30 M μ M ??

16. Line 381- RNAPs both.....RNAPs in both

17. Line 567: was carried out was carried out, repetition

18. *gdhA-4*.....*gdhA-4*

19. Line 1002-3 All ChIP-seq data in this 1003 figure are from a single, representative biological replicate. But originally how many replicates were there?

Katsuhiko Murakami, Penn State University

Reviewer #3:

Remarks to the Author:

Blombah et al. investigate the contribution of individual stages of the transcription cycle in transcriptional control in the archaeon *S. solfataricus*, and how they could serve as targets for regulation. The authors applied a combination of ChIP-seq and high-resolution ChIP-exo mapping to analyze in fine detail the genome-wide distribution of the basal transcription machinery, including RNA polymerase (RNAP) and the key initiation, elongation, and termination factors and to correlate it with changes in transcriptome upon a shift from exponential growth to stress growth conditions. The results implicate the promoter-proximal elongation step as the second major regulatory checkpoint (after recruitment of preinitiation complex (PIC) to promoters) determining promoter strength in archaea. The data presented also suggests that DNA duplex stability around the TSS may serve as an auxiliary regulator (both positive and negative) of transcriptional output, especially during oxidative stress.

The findings are novel and provide an important contribution to the field that warrants publication in Nature Communications. The results will be of interest to the research community studying archaeal transcription and its regulation in response to environmental cues. The main limitation of this work is that the key observations are based entirely on mathematical analysis of the correlations between ChIP-seq/ChIP-exo data (occupancy and escape index) and quantitative transcriptomics. While this approach is essential and proved to be very useful, additional in vitro transcription assays demonstrating promoter-proximal stalling and escape at susceptible promoters identified bioinformatically (e.g, *thsB*, *rps8a*, *dhg*, etc) would corroborate the findings and strengthen this study. I would also suggest performing the cluster analysis of transcription units that appear to be controlled at the stage of elongation escape; it could provide important information on metabolic pathways that depend on these regulatory circuits. Additionally, the manuscript contains several experimental discrepancies that should be resolved (see below).

1. Line 51. Use ref 52 instead of citation "Sanders et al 2020".
2. Introduction, last paragraph. To help readers who are not familiar with the subject, I suggest adding brief information about crenarchaeon *Sulfolobus solfataricus* and explain the reason for using this organism for the study.
3. Lines 88-90, and Figure 1. The authors should explain how they arrived at selecting for analysis such a small set (298 TUs) out of 2229 *S. solfataricus* TUs known from transcriptome analysis. What criteria were used for this selection?
4. Lines 95-98, 111-126, and Figure 1b. How far downstream and upstream of TSS does the ChIP-exo footprint signal for the PIC (RNAP, TFB, and TFE) extend on the template and non-template strands? It appears that it is not limited to 100 bp around TSS in both directions. This observation is very interesting, and yet it was not investigated any further and was discussed very briefly. The DNA scrunching should not extend beyond the promoter-proximal region (~30-50 bp). Could this signal result from the crosslink with archaeal histone-like proteins and nucleosomal DNA wrapped around the PIC?
5. Lines 102-104. The statement is a reasonable assumption but not a fact. The authors did not present ChIP-seq data for TBP.
6. Line 122. The authors probably mean to say "reconstituted", not "recombinant".
7. Lines 136-138, and Supplementary Figure 2. The TFB/RNAP occupancy scatter plots shown in Suppl. Fig. 2 are based on 454 promoters. The criteria used for promoter selection should be explained.
8. Lines 140-142, and Figure 2. Does Fig. 2 represent ChIP-seq or ChIP-exo data? The heterogeneity of RNAP distribution is not immediately evident from Fig. 2a. Perhaps cluster analysis would help? Also, the reasons for selecting such a small set of TU (212) should be explained.
9. Lines 170-172 and Figure 3ab. The conclusion that RNAP accumulation in the promoter-proximal region reflects reduced TEC escape into productive transcription is based on an analysis of only 212 TU. What about other TUs? How many TUs were used for classification? Also, the order of Spt4/5 vs Elf1 recruitment does not directly follow from Fig 3ab. The fact that EI values (higher occupancy at the promoter-proximal region) for Spt4/5 are higher than that of Elf1 may reflect more extensive contacts or stronger interactions with DNA and, therefore, more efficient crosslink.
10. Lines 200-202, and Figure 2a-d. The accumulation of aCPSF1 at the promoter-proximal region is evident from the heatmap and aggregated plot of Fig. 2a for a limited set of 212 TUs. Fig. 2b-d shows high occupancy at TSS and 3'-end of only three selected TUs. Is aCPSF1 observed at any TUs other than selected 212 TUs?
11. Lines 202-206, and Supplementary Figure 5. Contrary to the authors' statement, occupancies for RNAP and aCpsf1 at 41 mRNA 3'-ends are similar (unlike occupancies for Spt4/5 and Elf1!).
12. Lines 207-214, and Figure 4ab. Figure 4a is very confusing. The Y-axis is denoted as a centered log₂ signal of the ratio of aCPSF1 IP/Input. Thus, the aCPSF1's occupancy at TSS for all aggregated TUs is already low (the log₂ values are between 0 and 0.5, which corresponds to 1-1.4-fold enrichment over input DNA) and it further decreases at regions upstream and downstream of TSS (the log₂ values are negative, indicating that aCPSF1 is depleted relative to input DNA). Also, it's unclear what data were used to build the boxplots of Fig. 4b? Comparative occupancy plots of aCPSF1, RNAP, and spt4/5 for low and high EI should be presented.
13. Lines 215-225, Figure 4cd, and Supplementary Figure 6. The observed anticorrelations between relative aCPSF1 load and Spt4/5-TEC EI, especially mRNA expression levels, are relatively weak (Spearman's $r = -0.54$ and $r = -0.47$, respectively). Perhaps showing examples of specific TUs with strong correlation (in Supplementary data) would help support the stated conclusion (lines 222-225).
14. Lines 235-238, and Supplementary Figure 8. TFE α is not depleted from PICs under oxidative stress, whereas TFE β is (see Supl. Fig. 7). Yet, the scatter plots show a correlated enrichment for TFE α and TFE β . Authors should explain this discrepancy. Also, the top panels (peak overlaps) of Suppl Fig. 8 should be described in the Figure legend.
15. Lines 239-241. How many TUs out of 2229 TUs with established TSS are transcriptionally active (defined as high, medium, and low)? Which TU sets are used in plots shown in Figures 1-6, and why are the numbers of TUs different? Data filtering and selection of TU sets should be described in more

detail!

16. Figure 5b. The heatmap shown in Fig. 5b is puzzling; the data appear to be contradictory and difficult to understand. Authors state that changes in TEC escape (for Elf1 and Spt4/5 EI) positively correlate with changes in the transcriptome and TFB occupancy. Yet, changes in RNA output correlate poorly with changes in RNAP escape efficiency and TFE β occupancy. This conclusion seems illogical and requires an explanation. Also, the authors propose that the observed heterogeneity in TFE β promoter occupancy (relative to TFB) results from TFE β loss in stalled PICs and not because promoters have different affinities for TFE β under oxidative stress. Are there any data to support this hypothesis? What about changes in TFE α occupancy? Does it show a similar correlation pattern?

17. Lines 260-263. The statement seems incorrect. TFB accumulated at SSO8549 promoter to the same extent under both conditions. Do authors mean to say that there is no change in TFB enrichment in response to oxidative stress? Also, how does oxidative stress reduce RNAP escape? Are there any changes in escape efficiency at genes activated under oxidative stress? Perhaps, it would be more informative if authors perform cluster analysis of RNAP/factor occupancies and EIs at all TUs (upregulated, unchanged, and downregulated) to identify those that are regulated by changes in RNAP/factor accumulation at promoters (promoter-proximal pausing) and TEC escape.

18. Lines 281-283. Contrary to the authors' assertion, the observed changes in EIs for RNAP and SPT4/5 upon entering the stationary phase shown in Suppl. Fig. 11 are insignificant. The heatmap of correlated changes in TFB and TFE β occupancy and RNA output is in disagreement with the heatmap data for oxidative stress (Fig. 5b). The authors should provide more convincing data or remove this statement.

Our response to the reviewers highlighted in blue

REVIEWER

COMMENTS

Reviewer #1 (Remarks to the Author):

The authors describe an accumulation of promoter proximal transcription elongation complexes (TEC) in the archeon *Sulfolobus solfataricus*. Based on a number of high throughput sequencing experiments they argue that TEC escape into productive elongation is a rate limiting step in transcription that provides an additional level of regulation to archaean transcription. They show the recruitment of transcription elongation factors Spt4/5 and Elf1 along with termination factor aCSPF1 to the early TEC and suggest that the regulation might occur via blocking upstream initiating RNA polymerase (RNAP) or premature transcription termination.

This work provides a wealth of high throughput data and in general makes a compelling case for the functional importance of promoter proximal pausing in archaea. However, there is little explanation presented with respect to the mechanism of the demonstrated TEC accumulation.

We are grateful for reviewer-1's positive and encouraging comments about our work. As is often the case with novel discoveries arising from functional (multi-) genomics studies, the detailed structural and functional basis of the underlying mechanisms will be the subject of the next five to ten years of research. In the meantime, we do provide in-depth interpretation and suggest clear hypotheses about transcription regulation during promoter proximal elongation. This includes premature termination by the action of aCSPF1 and a negative feedback loop from slow TECs that interfere with PIC formation/turnover.

The Methods section describing the computational work could benefit from additional clarification, perhaps in the form of a supplemental text. This is especially true for the extensive raw data preprocessing steps that sometimes result in the discard of more than half of the reads (in particular, when adjusting the fragment size distribution). The inclusion of a github repository with some of the code used to perform analysis and generate figures is a welcome addition, but that too should be improved by additional documentation.

We aim at providing the best possible documentation for our methods including the code provided on github. For our ChIP-seq experiments we generated paired-end sequencing data. The pre-processing by sampling the reads in order to obtain identical DNA fragment size distributions for all ChIP-seq samples and chromatin

input improved the peak-calling for transcription initiation factors, a critical aspect of our analysis, as well as the normalisation of the data.

To improve the clarity of the manuscript, we have provided additional documentation in the Material and Methods section, and in the github repository including a readme.md file that assigns the R markdown scripts to the individual figures of the revised version.

The authors first use the ChIP-exo signal from the basal initiation factors TFB and TFE as well as from RNA polymerase to examine the footprint of the promoter initiation complex (PIC) around TSS and then invoke the ChIP-Seq data for RNAP and transcription elongation factors Spt4/5 and Elf1. The resulting profiles seem to suggest the accumulation of RNAP along with these two factors proximal to the promoter, but well outside of the PIC complex. In addition, there is a signal for the termination factor aCSPF1 with its peak corresponding to the apparent peak of RNAP. This suggests a possible interplay between premature termination and RNAP pausing.

Does RNAP backtracking play a role? Some discussion and relevant references would be appropriate.

In other systems, promoter-proximal pausing can lead to backtracking of TECs that can be rescued by transcript cleavage factors such as GreA and TFIIIS. This phenomenon has been reported for Sigma70-dependent pausing in bacteria and promoter-proximal pausing of RNAPII in eukaryotes and it is a possibility in archaea too. The *S. solfataricus* transcript cleavage factor TFS1 that is related to eukaryotic TFIIIS readily reactivates stalled TEC in vitro and this would likely counteract backtracking in vivo. However, despite several attempts, TFS1 cannot be ChIPed in *S. solfataricus* and was therefore not included in our occupancy analyses. We have added the following paragraph to the Discussion in the revised manuscript:

p.14, l.531

'Mechanisms underlying promoter-proximal TEC dynamics.

Besides premature termination, the promoter-proximal accumulation of TECs can be explained by slow elongation or pausing that might result in backtracking. Bacterial cleavage factor GreB facilitates the release of *E. coli* RNAP from Sigma70-dependent pause sites⁵⁸. Likewise, cleavage factor TFIIIS facilitates the release of promoter-proximally paused RNAP II in *Drosophila* and human cell lines^{59,60}. It is tempting to speculate that TFS, the archaeal TFIIIS homologue, might play a role in controlling promoter-proximal TEC dynamics in *Saccharolobus*. Unfortunately, our TFS ChIP-experiments were not successful and TFS association with promoter-proximal TECs remains to be tested.'

The addition of native elongating transcript sequencing (NET-Seq) data would help to elucidate both the functional status of RNAP and the underlying mechanism.

Whereas NET-seq experiments may be helpful, or not, this method has not been established in any archaeon or indeed hyperthermophile yet, and its technical development, application and data analysis would be beyond the scope of the current manuscript.

The authors further split transcriptional units into three groups with “low”, “intermediate” and “high” efficiency of TEC escape into productive elongation and quantify this by computing EI (Escape Index) as a log-transformed ratio of average RNA polymerase occupancy in TU body over the promoter region. The attempt to show the correlation between EI and the fraction of “short” RNAs synthesized as assayed by TSS-RNA seq is not very convincing. The statistical significance is apparent, but the size of the effect is modest and the biological relevance is questionable. This is further confirmed by the relatively low correlation between RNAP and Spt4/5 EI and RNA output in the comparison of these values during oxidative stress shown in Figure 5.

The TSS-RNA analysis (Figure 3) and the correlation matrix (Figure 5) examine two very different aspects of TEC escape. Firstly, the general correlation between RNAP escape and mRNA expression levels for cells in the exponential growth phase is *rather strong*. We had omitted these correlations for the sake of brevity. To address the reviewer’s comments, we added now an additional Supplementary Figure 4 with the direct pairwise correlations EI vs mRNA level.

RNA-seq measures steady-state mRNA levels and not the rate of RNA synthesis. Because of differences in mRNA half-life, our estimate is on the conservative side. We added the following sentence to the manuscript to describe this relation:

p.7, l.212: “Escape indices for all three proteins (RNAP, Spt4/5, Elf1) were strongly correlated with mRNA expression levels (Supplementary Figure 4).”

Regarding the TSS-RNA analysis, there are a few processes that could affect the levels of short TSS-RNAs in the cell: (i) nascent RNAs associated with promoter-proximal TECs, (ii) RNAs resulting from promoter-proximal transcription termination, (iii) RNA-turnover of longer, mature RNAs for example by 3’->5’ degradation by the archaeal exosome complex. (i) and (ii) are both related to RNAP escape. RNA turnover would be the only alternative explanation questioning the biological relevance of our findings. Crucially, mature RNA levels (estimated from standard RNA-seq) show a significantly weaker correlation with TSS-RNA levels arguing strongly for a direct,

biologically relevant link between occupancy of TECs at the promoter (ChIP-seq) and TSS-RNA levels (see Supplementary Figure 6).

The size of the effect is certainly underestimated due to the technical challenge to isolate, amplify, and sequence these short RNAs requiring multiple enzymatic steps including the necessary capping and isolation of capped RNAs. The nature and the level of bias that this protocol introduces is difficult to estimate, but the resulting correlation of Spearman's $r = 0.61$ between TSS-RNA levels and Spt4/5 promoter occupancy (not EI!) shown in Figure 3c strongly indicates both a statistically significant *and* biological relevant link between TSS-RNA levels and occupancy of TECs at the promoter (ChIP-seq).

The correlation matrix in Figure 5 tests the contribution of RNAP escape to the *gene-specific regulation* of transcription in contrast to a (possibly static) effect on promoter strength. As reviewer 1 pointed out, the correlation between RNA level changes and Spt4/5 or Elf1 escape indices is modest, but statistically significant. This indicates that, at least to some level, TEC escape can be tuned for gene-specific regulation of transcription. This particular analysis does not address any global effects on transcription attenuation in response to stress. Regarding the modest correlation, it is noteworthy that changes in TFB occupancy for the same set of genes do show a similarly modest correlation with changes in RNA levels. TFB as essential initiation factor is of unquestionable biological relevance for RNA synthesis. Possible reasons for these modest correlations include additional regulatory mechanisms such as RNA degradation in response to oxidative stress.

The purpose of the proposed multiple regression statistical model is not clearly explained. Was it meant as a predictive model? How was it trained and how was its performance validated? Was it meant to select the variables that explain most of the variance in the data?

We used generalised linear models for a multiple regression analysis. This is not a predictive model, but a mean to investigate the quantitative relationships between different factors beyond *pairwise* Spearman's rank correlations that have limited sensitivity. The analysis revealed an interesting new aspect. While both TFE β and TFB show stronger accumulation on promoters with low TEC escape (Figure 5), the multiple regression analysis shows that during oxidative stress (with TFE β levels depleted), promoters with low TEC escape show significantly weaker accumulation of TFE β relative to TFB. Thus, this analysis connects PIC dynamics with TEC escape in support of our negative feedback model for TEC escape regulation of transcription.

Minor:

Line 448: "We generated paired-end HiSeq data from our ChIP-Seq data..." - this obviously a typographical error.

Corrected

Line 449: "Reads were trimmed to 50 nt..." - trimmed how? On the 5' ends? What was the original read length?

The sentence has been corrected as follows:

p.18, l.646: 'Reads were trimmed from the 3'-ends to 50 nt read from 75 or 125 nt original read-length [...]'

Line 500: what is "filtering for consistency"?

We removed some annotated 'primary' TSSs which appeared to be internal TSSs within transcription units from a previous data set (Wurtzel et al. 2009). Our analysis pipeline included an independent filtering of internal TSSs based on TFB ChIP-seq data. We introduced and rephrased the beginning of the paragraph as follows to improve its clarity:

p.19, l.708: 'Because archaeal genomes are densely packed with TUs, we filtered the set of *S. solfataricus* TUs to ensure robust, TU-specific signal quantification of the ChIP-seq data. We manually curated a previously published map of 2229 TUs based on RNA-seq data.'

Figure 3 ab and Figure 4: The description is very confusing. If the data shown are from a single biological replicate, then what are the mean and standard deviation?

As these are aggregate plots, the mean and standard deviation refers to differences between the set of TUs. aCPSF1 aggregate profiles are now integrated into Figure 3ab (see reviewer 3 comments).

Reviewer #2 (Remarks to the Author):

Archaea constitute one of three domains of life, and their basal transcription system has been characterized as a hybrid of eukaryotic and bacterial transcription systems. Study and understanding of transcription regulation mechanisms in archaea is important to highlight the basic transcription regulation mechanisms across all three domains of life and give insights into functional evolution of multi-subunit polymerases transcription machinery.

The present study “Promoter-proximal elongation regulates transcription in archaea” by Blombach et al. focuses on how the transcription machinery is distributed along genes in the crenarchaeon *Sulfolobus solfataricus*. The authors utilized functional genomics approaches to reveal how transcription elongation complex (TEC) escape is rate-limiting for transcription read-out during exponential growth of *S. solfataricus*. By using ChIP-seq and ChIP-exo experiments, the authors showed a discrete footprint of RNA polymerase and different transcription initiation factors around the transcription start site (TSS), and their uniform assembly at promoter loci. They also reveal that the transition of RNAP by promoter escape into productive and processive elongation differs greatly across different transcriptional units. The results described in the manuscript establish that TEC escape is an important factor for determining promoter strength and transcript levels. The authors further compared the TEC escape during exponential growth and under oxidative stress conditions, surmising that the TEC escape is an integral part of the transcriptional stress response.

Overall, this is an extensive study by genomics approach investigating regulatory checkpoints in archaeal transcription and provides the basis for further in-vitro studies to decipher the underlying molecular mechanisms of TEC escape. Following points need to be addressed before considering for publication:

1. Line 26: Rephrase: “Transcription output is determined by the frequency of transcription initiation and premature termination”. It is unclear if “output” here is the net turnover or the overall rate.

In order to clarify this point, we replaced the term ‘transcription output’ with ‘rate of RNA synthesis’.

2. Authors showed RNAP is not effective in escaping into productive transcription in some TUs. But authors did not really explain why. What are the possible reasons? Do the authors think these results stay the same throughout different growth conditions?

The current manuscript describes a functional genomics study that among other insights identifies promoter proximal elongation as rate limiting step for mRNA levels

globally. The molecular mechanisms that could influence this process are likely manifold and will be subject for future investigation in the years to come, see also comments to reviewer-1. In the meantime, we do provide clear hypotheses about transcription regulation during promoter proximal elongation, such as the premature termination by the action of aCPSF1. And we agree with reviewer-2 that the factors influencing escape do change with changing growth conditions, as we provide evidence that oxidative stress leads to a loss of the significant association between escape and promoter-proximal bound aCPSF1.

3. Line 255-258: The results show that RNAP TEC escape is reduced in oxidative stress. But what could be the reason? Reason for accumulation of RNAP and Spt4/5 at promoters in oxidative stress conditions? How does an oxidative environment affect the canonical transcription pathway? Is this change could be archaea specific or omnipresent?

These are all important questions, unanswered in the archaea, but better understood in bacteria and eukaryotes. One intriguing fact is the depletion of TFE β in archaea, which likely results from the oxidative damage of a redox-sensitive cubane iron sulfur cluster (Blombach eLife 2015). However, our data do not show a strong link between TFE β recruitment and TEC escape. The increased accumulation of promoter-proximal TECs is reminiscent of the stabilisation of promoter-proximally paused human RNA polymerase II in response to oxidative stress by an unknown mechanism (Nilsson et al., 2017). More speculatively, oxidative stress could affect the DNA topology of the positively supercoiled hyperthermophilic genome, the chromatinization state of the genome or even alter the NTP substrate concentrations. But the field of archaeal molecular biology is still young, and there are no data to support these hypotheses in the literature, to the best of our knowledge. See also below.

4. Line 264: "The accumulation of TFB and other initiation factors": this sentence is not clear. Do authors mean that more than one TFB/other factors recruited to PIC and form non-functional PIC? Or a PIC is stuck at transcription initiation stage?

We work on the assumption that these PIC are 'stuck' as reviewer-2 puts it (see Supplementary Figure 13). We have modified this paragraph to clarify our reasoning.

p.11, l.406: 'The link between increased TFB accumulation and reduced TEC escape thus offers a possible mechanistic explanation how TEC escape can affect productive transcription. The accumulation of TFB at the promoter in ChIP-seq experiments could reflect a slower progression from the initial formation of ternary DNA-TBP-TFB complexes towards dissociation of TFB from RNAP during promoter clearance. To test whether RNAP recruitment to DNA-TBP-TFB is impaired...'

Authors should shed light on what are the key known changes under oxidative stress conditions. For e.g., NTP concentrations/less protein synthesis etc.?

As mentioned above, in archaea, frustratingly little is known about changes in the NTP concentrations or the impact of oxidative stress on translation. To clarify, we have introduced a sentence in the manuscript to clarify this point.

p.14 l.526: 'Changes in DNA topology as observed in *E. coli*⁵⁷, ribonucleotide concentrations, and DNA chromatinization are possibly contributing factors. However, our transcriptome data do not support strong induction of topoisomerase expression upon oxidative stress.'

5. Line 345: The authors suggest DNA duplex stability controls TEC escape. Can authors show the sequence comparison of some example promoters? How much difference in the stability or what percentage of G-C and A-T contents is considered as stable and unstable?

While GC-content is important, it is not the best determinant for DNA duplex stability. We predicted duplex stability using a nearest neighbour model based on the dinucleotide composition, rather than G-C content. As an indication, the 7-bp window encompassing positions -3 to +4 (relative to TSS) ranged from 14% to 71% GC content for the oxidative stress data set.

The correlation between escape indices and DNA duplex stability is calculated by using the deltaG estimates (their ranks to be more precise) without any categorization of stable and unstable sequences.

6. Through proofreading of manuscript is required for typos and article.

We carefully checked the manuscript for remaining typos.

7. *Sulfolobus solfataricus* has been called as *Saccharolobus solfataricus* since 2018. Authors doesn't have to use this new name but better to introduce it at the beginning of the manuscript.

We have changed the name throughout the manuscript. Unfortunately, the new NCBI taxonomy is not very consistent regarding the grouping of different *Sulfolobus* vs *Saccharolobus* species.

8. In the beginning of this manuscript, authors used a term "TEC escape" but it is not commonly used in transcription research field. Usually called as "promoter escape" and "RNAP escape" as authors used later in this manuscript. Authors may have a reason to call this process as "TEC escape". If so, define TEC escape.

The term 'promoter escape' is generally used for the process of RNAP clearing the promoter once the interactions between RNAP and DNA-bound initiation factors have been severed. One of our key findings is that another type of escape – from early inefficient elongation – makes a substantial contribution and sets the mRNA levels genome-wide. We have used the term 'RNAP escape' for both types of escape without discriminating between promoter clearance and processes affecting early elongation. The term 'TEC escape' emphasises this latter type of escape.

Minor comments:

9. Line 29: metazoans
10. Line 90, 98 Figure 1B.....1b
11. Line 114: Thymine basis..... bases
12. Line 120 – 122: The permanganate footprinting data is from this manuscript. "Figure 1F" citation is missing.
13. Line 192-194: references are missing
14. Figure 2h r 014..... 0.14
15. Supplementary figure S7: 30 M μ M ??
16. Line 381- RNAPs both.....RNAPs in both
17. Line 567: was carried out was carried out, repetition
18. gdha-4.....gdhA-4

We have introduced all corrections above.

19. Line 1002-3 All ChIP-seq data in this figure are from a single, representative biological replicate. But originally how many replicates were there?

Figure 4 has now been integrated into Figure 3 (see below). The sentence in the figure legend now states:

p.39, l.1298: 'Data are from a single representative of two biological replicates.'

Katsuhiko Murakami, Penn State University

Reviewer #3 (Remarks to the Author):

Blombach et al. investigate the contribution of individual stages of the transcription cycle in transcriptional control in the archaeon *S. solfataricus*, and how they could serve as targets for regulation. The authors applied a combination of ChIP-seq and high-resolution ChIP-exo mapping to analyze in fine detail the genome-wide distribution of the basal transcription machinery, including RNA polymerase (RNAP) and the key initiation, elongation, and termination factors and to correlate it with changes in transcriptome upon a shift from exponential growth to stress growth conditions. The results implicate the promoter-proximal elongation step as the second major regulatory checkpoint (after recruitment of preinitiation complex (PIC) to promoters) determining promoter strength in archaea. The data presented also suggests that DNA duplex stability around the TSS may serve as an auxiliary regulator (both positive and negative) of transcriptional output, especially during oxidative stress.

The findings are novel and provide an important contribution to the field that warrants publication in Nature Communications. The results will be of interest to the research community studying archaeal transcription and its regulation in response to environmental cues. The main limitation of this work is that the key observations are based entirely on mathematical analysis of the correlations between ChIP-seq/ChIP-exo data (occupancy and escape index) and quantitative transcriptomics. While this approach is essential and proved to be very useful, additional *in vitro* transcription assays demonstrating promoter-proximal stalling and escape at susceptible promoters identified bioinformatically (e.g, *thsB*, *rps8a*, *dhg*, etc.) would corroborate the findings and strengthen this study.

In vitro models for promoter-proximal elongation and pausing have the potential to shed further light into the mechanistic aspects of our functional genomics study. However, it is important to keep in mind, that standard *in vitro* models utilise artificially low ribonucleotide concentrations, or even the omission of a specific ribonucleotide (reviewed in Core and Adelman, 2019, Genes & Dev). Only very recently the Dylan Taatjes lab (Fant et al., 2020, Mol Cell) succeeded in establishing an *in vitro* transcription assay that does not depend on limiting ribonucleotide concentrations. These studies are all limited to very small sets of model promoters, and this holds also true for Sigma70-dependent pausing in *E. coli*. Thus, it is fair to say that the development of an archaeal *in vitro* model for promoter-proximal regulation is very challenging.

Despite these challenges, we have now developed an *in vitro* transcription protocol for synchronised *in vitro* transcription that allows us to measure promoter-proximal transcription elongation. Our system is based on cell lysates, thus all required factors are present, and we included an antisense oligonucleotide capture affinity purification

step of the generated transcripts which makes it possible to use the native sequence context (rather than e. g. C-less cassettes). We applied this system to all four promoters shown in Figure 2 as suggested by the reviewer. The resulting *in vitro* transcription method and the results are now presented in the new Figure 4. Importantly, the CRISPR C promoter with low EI shows a significant accumulation of short transcripts likely due to pausing during early elongation.

p.9, l.294: 'The crispr C promoter shows pausing in the promoter-proximal region in vitro

To test whether promoter-proximal pausing of RNAP can be observed in vitro for low TEC escape promoters, we developed a synchronised in vitro transcription assay to monitor promoter-proximal transcription elongation dynamics. We used *S. solfataricus* cell lysates generated from cells in exponential growth phase to provide a full set of auxiliary factors.

To generate templates for in vitro transcription, we inserted target promoter regions encompassing -50 to +100 relative to the TSS into plasmids that were subsequently linearised downstream of the insert to allow for run-off transcripts of 115 nt length. Synchronisation of transcription was achieved by a transcriptionally inhibitory variant of TFB termed TFBC that comprises only the C-terminal cyclin fold domains⁶⁰. Pre-formed PICs are able to initiate a single round of transcription, but subsequent PIC assembly and transcription re-initiation is blocked by an excess of TFBC outcompeting the inherent TFB for recruitment to TBP-bound promoters (Figure 4a). The generated transcripts were affinity purified using immobilised 25 nt antisense oligonucleotides. We tested the assay on two promoters showing high TEC escape (*thsB* and *rps8e*) and two promoters showing low TEC escape (*dhg-1* and CRISPR C), the same promoters with ChIP-seq profiles depicted in Figure 2b-e. All four promoters gave rise to run-off transcripts within 30s under the experimental conditions (Figure 4b). Notably, the CRISPR C promoter displayed some level of early, broad pausing at 30 to 40 nt transcript length, about the shortest length that is reliably detectable with the assay. Thus, the in vitro transcription data for the CRISPR C promoter are in line with early pausing of TECs in the promoter-proximal region. The absence of a corresponding pausing pattern for the *dhg-1* promoter may suggest that it is difficult to establish the proper context such as chromatinization of the DNA templates that reflects the in vivo situation.'

I would also suggest performing the cluster analysis of transcription units that appear to be controlled at the stage of elongation escape; it could provide important information on metabolic pathways that depend on these regulatory circuits.

The data filtering process that is required to obtain meaningful measurements of TEC escape *limits* the information that can be obtained from a cluster analysis approach. Our data set contains many TUs encoding house-keeping genes that are highly

expressed (such as *rps8e* and *thsB*) and these TUs do show generally high TEC escape as one might expect. We did check manually all TUs with low TEC escape but did not find any overarching biological function of low escape TUs.

Additionally, the manuscript contains several experimental discrepancies that should be resolved (see below).

1. Line 51. Use ref 52 instead of citation "Sanders et al 2020".

corrected

2. Introduction, last paragraph. To help readers who are not familiar with the subject, I suggest adding brief information about crenarchaeon *Sulfolobus solfataricus* and explain the reason for using this organism for the study.

We added a short paragraph.

p.3, l.76: 'The crenarchaeon *Saccharolobus solfataricus* (formerly *Sulfolobus*) is a well-established model organism for archaeal transcription (e.g. Qureshi1997, Hirata2008, Blombach2015, Sheppard2016, Fouqueau2017). Importantly, *S. solfataricus* harbours the full repertoire of known archaeal transcription elongation factors including Elf1 making it a good choice to investigate archaeal transcription regulation beyond the initiation stage.

3. Lines 88-90, and Figure 1. The authors should explain how they arrived at selecting for analysis such a small set (298 TUs) out of 2229 *S. solfataricus* TUs known from transcriptome analysis. What criteria were used for this selection?

We added a short paragraph to explain the TU selection process. The detailed procedure is described in the Methods section which we also tried to improve in terms of clarity as suggested by reviewer 1.

In brief, we applied filtering criteria that we feel are required to obtain reliable quantifications of ChIP-seq occupancies attributed to a specific TU. These criteria were applied to both biological replicates.

- A 20 reads minimum coverage for chromatin input samples to ensure reliable input normalisation of the ChIP-seq data
- Biunique correspondence between initiation factor peaks (TFB and TFE β) and the TSS of TUs to ensure that we can attribute the initiation factor ChIP-seq occupancy to specific promoters. This is critical to deal with frequent closely-spaced pairs of promoters for divergent TUs

- In, addition, we required a 1.5x increase in Spt4/5 signal in order to ensure that the promoter-proximal Spt4/5 signal can be attributed to the TU
- In order to deal with frequent TU-internal promoters, we checked for internal TFB peaks. Because low TEC escape TUs show low RNAP occupancy within the TU body, RNAPs recruited TU-internal promoters will skew the estimate of RNAP escape significantly
- To measure TU-internal RNAP occupancy by ChIP-seq, we also required a minimum TU length of 500 bp to obtain a large enough window (+250 to +500) that is well isolated from signal at the promoter region. For the ChIP-exo data analysis focussing on promoter occupancy exclusively, we shortened the minimum TU length to 100 nt

The relatively small set of TUs is a 'necessary evil' of this type of study. However, the filtering is necessary to obtain robust EI measurements. We optimised the TU filtering process extensively to obtain a good compromise between the number of TUs investigated and the robustness of measurements obtained from them.

p.4., l.106: 'We calculated aggregate profiles of ChIP-exo data for a set of 298 TUs with mapped TSS. These TUs were selected from a total set of 1054 mapped TSSs based on the corresponding ChIP-seq data for TFB and TFE β to include only TUs displaying TFB and TFE β occupancy as well as to exclude TUs with problematic regions for the mapping of sequencing reads (see methods).'

We have added the scripts for TU selection to the github repository.

4. Lines 95-98, 111-126, and Figure 1b. How far downstream and upstream of TSS does the ChIP-exo footprint signal for the PIC (RNAP, TFB, and TFE) extends on the template and non-template strands? It appears that it is not limited to 100 bp around TSS in both directions. This observation is very interesting, and yet it was not investigated any further and was discussed very briefly. The DNA scrunching should not extend beyond the promoter-proximal region (~30-50 bp). Could this signal result from the crosslink with archaeal histone-like proteins and nucleosomal DNA wrapped around the PIC?

We estimate from the comparison of the different ChIP-exo profiles (RNAP, TFB, TFE β) that the profiles are dominated protein-protein cross-links within the PIC – and could include additional uncharacterised proteins. DNA-wrapping or cross-linking to chromatin proteins is a likely explanation for the broad profiles as mentioned in the DISCUSSION section. We now extended the above-mentioned part of the RESULTS section as well:

p.5, l.153: 'suggest that additional, yet uncharacterised components associate with the PIC in the cell such as chromatin proteins or PICs interact with downstream DNA.'

5. Lines 102-104. The statement is a reasonable assumption but not a fact. The authors did not present ChIP-seq data for TBP.

We toned down the statement by replacing "show" with "suggest".

6. Line 122. The authors probably mean to say "reconstituted", not "recombinant".

corrected

7. Lines 136-138, and Supplementary Figure 2. The TFB/RNAP occupancy scatter plots shown in Suppl. Fig. 2 are based on 454 promoters. The criteria used for promoter selection should be explained.

We added the selection criteria to the Figure legend:

'454 promoters with unambiguous assignment of TFB and TFE β peaks to mapped TSSs or start codons (if mapped TSS was not available) and 100 bp minimal TU length were included.'

8. Lines 140-142, and Figure 2. Does Fig. 2 represent ChIP-seq or ChIP-exo data? The heterogeneity of RNAP distribution is not immediately evident from Fig. 2a. Perhaps cluster analysis would help? Also, the reasons for selecting such a small set of TU (212) should be explained.

The results are ChIP-seq data and this information has now been added to the figure legend. A cluster analysis would be appropriate if we could identify distinct subpopulations of low and high escape TUs. However, the escape index calculations suggest a continuum (see e. g. Figure S3).

9. Lines 170-172 and Figure 3ab. The conclusion that RNAP accumulation in the promoter-proximal region reflects reduced TEC escape into productive transcription is based on an analysis of only 212 TU. What about other TUs? How many TUs were used for classification?

We explained above why we consider it necessary to apply stringent filtering of the TUs to be analysed above. It is difficult to estimate to what extent the data filtering process is biased against or in favour of TUs with low TEC escape. On the one hand, the accumulation of initiation factors at TUs with low TEC escape will indeed increase the chance of detecting TFB and TFE β peaks for these TUs and thus including them

in the data analysis. On the other hand, TEC escape estimates for weakly expressed TUs the ChIP-seq signal is increasingly dominated by noise preventing reliable TEC escape measurements.

Concerning Figure 3a/b, we classified the high and low escape TUs from this set of 212 TUs. We altered the legend for Figure 3 to clarify this point:

'Recruitment of elongation factor Spt4/5 precedes Elf1 and aCPSF1 recruitment to the TEC. Aggregate plots for TUs with high (mean EI RNAP > -1, n= 58) (a) and low escape indices (< -2.5, n= 41) from the set of 212 TUs analysed in Figure 2.'

Also, the order of Spt4/5 vs Elf1 recruitment does not directly follow from Fig 3a/b. The fact that EI values (higher occupancy at the promoter-proximal region) for Spt4/5 are higher than that of Elf1 may reflect more extensive contacts or stronger interactions with DNA and, therefore, more efficient crosslink.

The hypothesis proposed by the reviewer implies that Spt4/5 contacts with DNA are altered during the transition from early into productive elongation. The tight association of Spt4/5 with RNAP and the nucleic acid scaffold in the TEC does not support the notion of such structural flexibility (Ehara et al., 2017 Science). More importantly, we identified several low escape TUs that do recruit Spt4/5 to significantly higher levels than Elf1 and aCPSF1. This finding is incompatible with concomitant recruitment of all three factors, but it can be readily explained by TECs stalling at an earlier stage of elongation prior to Elf1 and aCPSF1 recruitment in line with the recruitment cascade that we proposed. We have added an additional sentence and a new Supplementary Figure 5 depicting two examples of TUs showing these characteristics:

p.7, l.217: "Consecutive recruitment of elongation factors is consistent with the observation that some TUs with low RNAP escape showed a much stronger Spt4/5 recruitment compared to Elf1 suggesting that TECs stall at an earlier stage, prior to Elf1 recruitment, on these TUs (Supplementary Figure 5)."

10. Lines 200-202, and Figure 2a-d. The accumulation of aCPSF1 at the promoter-proximal region is evident from the heatmap and aggregated plot of Fig. 2a for a limited set of 212 TUs. Fig. 2b-d shows high occupancy at TSS and 3'-end of only three selected TUs. Is aCPSF1 observed at any TUs other than selected 212 TUs?

aCPSF1 was not part of the TU selection criteria. Peak calling retrieved a total of 1633 aCPSF1 peaks after IDR-filtering. This is on similar scale as the 1657 TFB peaks discovered using the same criteria. Therefore, it is reasonable to assume that aCPSF1 is recruited to the vast majority of TUs similar to TFB. We corroborated this by looking

at TSSs of TUs in divergent orientation that should be void of any termination-associated aCPSF1 signal.

Concerning Figure 2b-d, *rps8e* and *dhg-1* both have 3'-ends overlapping with promoters as evident from TFB and TFE occupancies. Thus, the aCPSF1 peaks at the 3'-end region are very likely to come from aCPSF1 associated with these downstream promoters. Only for *thsB* there is an unambiguous aCPSF1 peak at the 3'-end. The results presented in Supplementary Figure 8 reflect this issue.

We added the following sentences to the manuscript:

p.8, l.254: 'We also analysed aCPSF1 association with predicted TSSs from pairs of TUs in divergent orientation. These TSSs are thus isolated from termination sites at the 3'-end of TUs. aCPSF1 peaks showed strong association with these TSSs similar to initiation factor TFB that we used as control in line with the results above (Supplementary Figure 7).'

11. Lines 202-206, and Supplementary Figure 5. Contrary to the authors' statement, occupancies for RNAP and aCpsf1 at 41 mRNA 3'-ends are similar (unlike occupancies for Spt4/5 and Elf1!).

The RNAP and aCPSF1 profiles are *similar* and our sentence was meant to say exactly that. The important point is that aCPSF1 does not form *distinct peaks* in the termination region - in contrast to its enrichment in the promoter region!

We modified the sentence for the sake of clarity:

p.8, l.260: 'Instead we observed a decrease in occupancy of aCPSF1 together with RNAP downstream of the predicted mRNA 3'-ends, and only in some cases well defined CPSF1 peaks'

12. Lines 207-214, and Figure 4ab. Figure 4a is very confusing. The Y-axis is denoted as a centered log₂ signal of the ratio of aCPSF1 IP/Input. Thus, the aCPSF1's occupancy at TSS for all aggregated TUs is already low (the log₂ values are between 0 and 0.5, which corresponds to 1-1.4-fold enrichment over input DNA) and it further decreases at regions upstream and downstream of TSS (the log₂ values are negative, indicating that aCPSF1 is depleted relative to input DNA).

With 'centered' we refer to the standard procedure of subtracting the mean value from the data. This explains the apparent low values. We prefer our method over standard aggregate profiles that just average the data. Importantly, the shape of the aggregate profile is not influenced by the centering procedure. It just serves to provide a meaningful standard deviation that solely reflects heterogeneity in shape

of individual occupancy profiles, but not heterogeneity in overall occupancy arising from weak and strong promoters.

We slightly changed this normalisation procedure by subtracting now RNAP_{Bd} values for all factors in order to accommodate also aCPSF1 aggregate plots into this figure (see below).

We improved the axis label for clarity. It states now:
'mean $\log_2(\text{IP}/\text{input}) - \log_2(\text{RNAP}_{\text{Bd}})$ '

Also, it's unclear what data were used to build the boxplots of Fig. 4b? Comparative occupancy plots of aCPSF1, RNAP, and spt4/5 for low and high EI should be presented.

Following the reviewer's suggestion to improve the comparison between aCPSF1 and Spt4/5, we integrated aCPSF1 aggregate profiles into Figure 3 (along the other panels of former Figure 4). This change made the boxplots redundant in our view and we have removed them.

13. Lines 215-225, Figure 4cd, and Supplementary Figure 6. The observed anticorrelations between relative aCPSF1 load and Spt4/5-TEC EI, especially mRNA expression levels, are relatively weak (Spearman's $r=-0.54$ and $r=-0.47$, respectively). Perhaps showing examples of specific TUs with strong correlation (in Supplementary data) would help support the stated conclusion (lines 222-225).

Generally, we do not expect very strong correlations with mRNA levels because promoter-proximal regulation is only one of several regulatory mechanisms including modulated RNAP recruitment during transcription initiation, and RNA decay. Furthermore, aCPSF1-mediated termination appears to be only one of at two different mechanisms of promoter-proximal regulation that affect TEC escape (Spt4/5 EI) and mRNA levels (the other being feedback affecting transcription initiation). Finally, the correlation is weaker compared to the corresponding values for Elf1 because a small subset of promoters appears to stall at an earlier Spt4/5-bound stage of elongation prior to recruitment of Elf1 and aCPSF1. These genes appear as outliers in the lower left corner of the scatter plot for Spt5 in Supplementary Figure 9 (formerly 6), but not in the corresponding scatter plot for Elf1 in Figure 4.

The examples shown in Figure 2 to reflect the basic relationship between high aCPSF1 recruitment and mRNA expression because of the association of aCPSF1 with lower TEC escape. However, because aCPSF1-mediated termination is part of TEC escape regulation and the triangular relation between aCPSF1 recruitment, TEC escape and mRNA expression is difficult to represent in such exemplary single TU

plots and cannot be assessed in pairwise correlations. For this reason, we conducted the multiple regression analysis in Supplementary Figure 16 (formerly 14).

14. Lines 235-238, and Supplementary Figure 8. TFE α is not depleted from PICs under oxidative stress, whereas TFE β is (see Supl. Fig. 7). Yet, the scatter plots show a correlated enrichment for TFE α and TFE β . Authors should explain this discrepancy. Also, the top panels (peak overlaps) of Suppl Fig. 8 should be described in the Figure legend.

This is a misunderstanding. While TFE β is depleted from *the cytoplasm* in response to oxidative stress as detected by immunoblotting, TFE α is not. However, both TFE subunits are depleted from promoter-bound PICs as shown in the ChIP-seq analyses. This shows that TFE α recruitment to the PIC depends on TFE β , similar to the homologous RPC62/39 subcomplex in human RNA polymerase III. The respective sentence was modified as follows:

p.10, l.332: 'TFE β depletion coincided with globally decreased promoter occupancies for both TFE α and β subunits in ChIP-seq suggesting that TFE α recruitment to the promoter is TFE β -dependent (Supplementary Figure 11).'

In addition, we added the requested information to the figure legend (now Suppl Fig. 9).

'The upper panels depict Venn diagrams of TFE α and TFB peaks that overlap with TFE β peaks (as defined by maximum 40 bp distance between peaks) under exponential growth (left panel) and oxidative stress conditions (right panel). The total peak number for each factor is stated in brackets.'

15. Lines 239-241. How many TUs out of 2229 TUs with established TSS are transcriptionally active (defined as high, medium, and low)? Which TU sets are used in plots shown in Figures 1-6, and why are the numbers of TUs different? Data filtering and selection of TU sets should be described in more detail!

The phrase 'transcriptionally active' does not sufficiently describe our data filtering and we have removed it from the manuscript. The point concerning the description of the data filtering process has been raised multiple times by different reviewers and we have made several changes throughout the manuscript to provide the requested information. Here, we applied the same basic filtering criteria to both exponential and oxidative stress data sets to determine reliable estimates for TEC escape. We then used the intersection of TUs – the ones that are shared between the two data sets - to compare TEC escape under both conditions.

'Transcriptionally active' in the sense of observed mRNA level was not part of filtering process.

We modified the section referred to above as follows to provide a more accurate description:

p.10, l.336: 'To understand how oxidative stress affects TEC escape, we generated ChIP-seq data for RNAP, initiation and elongation factors and we applied the same data filtering as for exponential growth phase data to obtain a set of 118 TUs with EI estimates. We then analysed the intersection of 71 TUs from both exponential growth and oxidative stress data sets to compare TEC escape between the two conditions. TUs displaying high TEC escape under exponential growth conditions showed overall reduced RNAP and Spt4/5 escape in response to oxidative stress (Figure 5a).'

16. Figure 5b. The heatmap shown in Fig. 5b is puzzling; the data appear to be contradictory and difficult to understand. Authors state that changes in TEC escape (for Elf1 and Spt4/5 EI) positively correlate with changes in the transcriptome and TFB occupancy. Yet, changes in RNA output correlate poorly with changes in RNAP escape efficiency and TFE β occupancy. This conclusion seems illogical and requires an explanation.

This is an important point. RNAP EI measurements also reflect accumulation of PICs when TEC escape is reduced. We conclude this, because reduced TEC escape leads to higher accumulation of TFB at the promoter (Figure 5) and changes in TFB and RNAP promoter occupancy remain highly correlated as determined by ChIP-exo (see Supplementary Figure 13). At this stage, we can only speculate whether this may explain the weaker correlation. We added the following section to the manuscript:

p.11, l.417: 'The finding that PICs accumulate to higher levels when TEC escape is reduced indicates that changes in RNAP escape indices reflect both cause and effect of TEC escape regulation. Curiously, changes in RNAP EIs do show a weaker correlation to transcriptome changes compared to Spt4/5 and Elf1.'

Also, the authors propose that the observed heterogeneity in TFE β promoter occupancy (relative to TFB) results from TFE β loss in stalled PICs and not because promoters have different affinities for TFE β under oxidative stress. Are there any data to support this hypothesis? What about changes in TFE α occupancy? Does it show a similar correlation pattern?

TFE β can only be recruited to the PIC as heterodimeric complex with TFE α , and the results presented in Supplementary Figure 11 suggest that likewise TFE β is required for recruitment of TFE α to PICs because TFE α promoter occupancy *remains* highly correlated with TFE β occupancy under oxidative stress conditions. We originally

intuitively assumed that TFE β recruitment would be limiting for gene expression during oxidative stress conditions, thus showing a stronger correlation to mRNA levels than TFB, however, our results do not support this hypothesis.

We therefore speculate and propose an alternative hypothesis: loss of TFE β from PICs.

17. Lines 260-263. The statement seems incorrect. TFB accumulated at SSO8549 promoter to the same extent under both conditions. Do authors mean to say that there is no change in TFB enrichment in response to oxidative stress?

We have corrected the sentence. It states now:
'[...] no significant changes in TFB accumulation'.

Also, how does oxidative stress reduce RNAP escape? Are there any changes in escape efficiency at genes activated under oxidative stress? Perhaps, it would be more informative if authors perform cluster analysis of RNAP/factor occupancies and EIs at all TUs (upregulated, unchanged, and downregulated) to identify those that are regulated by changes in RNAP/factor accumulation at promoters (promoter-proximal pausing) and TEC escape.

As discussed above, future functional studies will explore the structural and mechanistic basis of this phenomenon. Many TUs that are activated in the oxidative stress response such as *dps-1* show low transcription activity under exponential growth conditions which makes it impossible to assess changes in RNAP escape for these TUs. In all likelihood, gene specific transcription activators or repressors are involved in the dramatic induction of stress response genes – beyond the global regulation that is the subject of our study. To the best of our knowledge, there is no good way to deal with the absence of EI measurements for these TUs in a cluster analysis.

18. Lines 281-283. Contrary to the authors' assertion, the observed changes in EIs for RNAP and SPt4/5 upon entering the stationary phase shown in Suppl. Fig. 11 are insignificant.

The heatmap of correlated changes in TFB and TFE β occupancy and RNA output is in disagreement with the heatmap data for oxidative stress (Fig. 5b). The authors should provide more convincing data or remove this statement.

The pair-wise Wilcoxon rank sum test that we originally used inflates the p-value because it treats the 74 TUs as independent observations as in the case of the oxidative stress comparison in Figure 5a. We changed Figure 5a to scatter plot format

with Welch's t-test comparison for individual TUs as a more appropriate way of statistical testing.

We do agree that the heatmap shows apparent differences between oxidative stress and stationary phase regarding how changes in TFE β correlate with changes in mRNA levels. The central finding, that the EI changes positively correlate with change in mRNA levels, holds true for both oxidative stress and stationary phase. To reduce the complexity of the manuscript, we followed the reviewer's suggestion and removed the stationary phase data.

Reviewers' Comments:

Reviewer #1:

Remarks to the Author:

I have read the revised manuscript and rebuttal and I am fully satisfied with their response.

Reviewer #2:

Remarks to the Author:

Authors addressed all critiques from this reviewer on a revised manuscript. This reviewer recommends to publish this work.

Reviewer #3:

Remarks to the Author:

The revised manuscript is a great improvement. The development of an in vitro transcription protocol and demonstration of promoter-proximal pausing for two promoters is a big plus! The authors addressed most of my questions and concerns. There are still few minor errors that should be corrected.

1) Page 27, line 933. "50 μ M rNTPs (or as indicated) were provided..." The concentration of NTPs (50 μ M) seems very low for efficient transcription initiation. No other concentrations of NTPs are mentioned in the text. Please verify. Also, how were the transcripts radiolabeled (see Figure 4B legend)?

2) Page 27, lines 935-937. Please indicate the incubation time for Proteinase K treatment.

3) Page 27, line 949. Is it really 19 M EDTA? Indicate the correct concentration of EDTA.

4) Page 28, lines 950-952, and Figure 4. The authors should describe the labeling of transcripts before PAGE analysis and visualization by phosphorimager.

5) In the authors' response to Reviewer 2, line # 526 on p. 14 and ref. #57 are indicated incorrectly. It should be line #506 and Ref #64. Also, the authors should provide a full description for Ref #64.

REVIEWERS' COMMENTS

Reviewer #1 (Remarks to the Author):

I have read the revised manuscript and rebuttal and I am fully satisfied with their response.

We are happy to hear that and thank the reviewer for the constructive feedback.

Reviewer #2 (Remarks to the Author):

Authors addressed all critiques from this reviewer on a revised manuscript. This reviewer recommends to publish this work.

Again, we thank the reviewer for the constructive feedback.

Reviewer #3 (Remarks to the Author):

The revised manuscript is a great improvement. The development of an in vitro transcription protocol and demonstration of promoter-proximal pausing for two promoters is a big plus! The authors addressed most of my questions and concerns. There are still few minor errors that should be corrected.

We also thank this reviewer for the excellent feedback and the careful look at the new in vitro transcription experiments in our manuscript. Where necessary, we have modified the text in order to clarify the protocol for in vitro transcription.

1) Page 27, line 933. "50 μ M rNTPs (or as indicated) were provided..." The concentration of NTPs (50 μ M) seems very low for efficient transcription initiation. No other concentrations of NTPs are mentioned in the text. Please verify. Also, how were the transcripts radiolabeled (see Figure 4B legend)?

The final concentration of rNTPs is indeed 50 μ M. The reviewer is right to point out that this is certainly below physiological levels. However, the reduced rNTP concentration is required in order to facilitate the time course experiments.

Concerning the radiolabelling of transcripts, we included [α - 32 P]-UTP in the rNTP mix. We modified the text as follows to clarify this point:

"50 μ M rNTPs including trace amounts of [α - 32 P]-UTP (Perkin Elmer) were provided to initiate transcription together with 5 μ M TFBC blocking further pre-initiation complex assembly."

and the legend for Figure 4B:

"after simultaneous addition of 50 μ M rNTPs including [α - 32 P]-UTP and TFBC."

2) Page 27, lines 935-937. Please indicate the incubation time for Proteinase K treatment.

The text has been modified to include this information:

"After 5 min incubation, samples were purified..."

3) Page 27, line 949. Is it really 19 M EDTA? Indicate the correct concentration of EDTA.

It is indeed 18 mM EDTA. This is a standard composition of formamide loading dyes for Urea PAGE.

4) Page 28, lines 950-952, and Figure 4. The authors should describe the labeling of transcripts before PAGE analysis and visualization by phosphorimager.

Please see above

5) In the authors' response to Reviewer 2, line # 526 on p. 14 and ref. #57 are indicated incorrectly. It should be line #506 and Ref #64. Also, the authors should provide a full description for Ref #64.

We apologise for this mistake. Ref 64 has now been corrected.